# Molecular characterization of Richter syndrome identifies de novo diffuse large B-cell lymphomas with poor prognosis

Julien Broséus [1,2,3,32] ✉, Sébastien Hergalant [2,32], Julia Vogt [4],
Eugen Tausch[1], Markus Kreuz[5], Anja Mottok[4], Christof Schneider[1],
Caroline Dartigeas[6], Damien Roos-Weil[7], Anne Quinquenel[8], Charline Moulin[9,10],
German Ott[11], Odile Blanchet [12], Cécile Tomowiak [13,14], Grégory Lazarian[15],
Pierre Rouyer[2], Emil Chteinberg[4], Stephan H. Bernhart[16], Olivier Tournilhac[17],
Guillaume Gauchotte[2,18], Sandra Lomazzi[19], Elise Chapiro [20,21],
Florence Nguyen-Khac [20,21], Céline Chery[2,22], Frédéric Davi[21,23],
Mathilde Hunault[24], Rémi Houlgatte[2], Andreas Rosenwald[25], Alain Delmer [8],
David Meyre[2], Marie-Christine Béné [26,27], Catherine Thieblemont[28],
Peter Lichter[29], Ole Ammerpohl[4], Jean-Louis Guéant [2,22], ICGC MMML-Seq
Consortium*, Romain Guièze[17], José Ignacio Martin-Subero [30,31],
Florence Cymbalista[15], Pierre Feugier[2,9,33], Reiner Siebert[4,33] &
Stephan Stilgenbauer [1,33] ✉

Richter syndrome (RS) is the transformation of chronic lymphocytic leukemia
(CLL) into aggressive lymphoma, most commonly diffuse large B-cell lym-
phoma (DLBCL). We characterize 58 primary human RS samples by genome-
wide DNA methylation and whole-transcriptome profiling. Our comprehensive
approach determines RS DNA methylation profile and unravels a CLL epige-
netic imprint, allowing CLL-RS clonal relationship assessment without the
need of the initial CLL tumor DNA. DNA methylation- and transcriptomic-
based classifiers were developed, and testing on landmark DLBCL datasets
identifies a poor-prognosis, activated B-cell-like DLBCL subset in 111/
1772 samples. The classification robustly identifies phenotypes very similar to
RS with a specific genomic profile, accounting for 4.3-8.3% of de novo DLBCLs.
In this work, RS multi-omics characterization determines oncogenic mechan-
isms, establishes a surrogate marker for CLL-RS clonal relationship, and pro-
vides a clinically relevant classifier for a subset of primary "RS-type DLBCL"
with unfavorable prognosis.

Chronic lymphocytic leukemia (CLL) is the most frequent leukemia in Western countries[1]. While generally considered an indolent B cell disease, CLL is in fact associated with a highly heterogeneous clinical course. CLLs are classified into two major molecular subtypes that differ in their degree of somatic hypermutations in the immunoglobulin heavy chain variable (IGHV) domains. IGHV-unmutated CLLs (U-CLL) are associated with an inferior prognosis than IGHV-mutated CLLs (M-CLL)[2]. CLL transformation into a more aggressive histology is

A full list of affiliations appears at the end of the paper. *A list of authors and their affiliations appears at the end of the paper.
✉e-mail: julien.broseus@univ-lorraine.fr; Stephan.Stilgenbauer@uniklinik-ulm.de

termed Richter syndrome (RS)[3]. Diffuse large B cell lymphoma (DLBCL) subtype accounts for 90–95% of RS cases. Around 80% of RS cases are IGHV-unmutated while the remainder are IGHV-mutated[4]. In contrast, most de novo DLBCL (from now on called DLBCL) are IGHV-mutated as they originate from germinal center (GC) or post-GC B cells. Based on gene expression patterns, different cell-of-origin (COO) derivations of DLBCL include GC B cell like (GCB) and activated B cell-like (ABC) DLBCL[5]. Recent genomic studies combining DNA and RNA sequencing extended DLBCL subtyping beyond COO[6–10], identifying DLBCL subgroups defined by their genomic alteration patterns and associated clinical courses, but a notable proportion remains unclassified[7,8,10]. Moreover, although studies have shown some extent of association of genetically defined groups with transcriptionally defined COO signatures, the transcriptome in its entirety is not fully used in current classifications.

As compared to other lymphoid malignancies, the availability of in vitro or in vivo models to study RS is limited[11–15], and therefore our current knowledge on RS biology remains incomplete. The few genomic studies attempting to decipher oncogenic mechanisms underlying RS described disabled DNA damage response and cell cycle control through *TP53* abnormalities and *CDKN2A* deletions, chronic B cell receptor (BCR) signaling, and NOTCH, MYC, and MAPK pathway deregulations[16–20]. A recent report using multiome and single cell approaches in sequential CLL-RS samples describes that the increased molecular complexity of RS does not seem to be the consequence of clonal evolution over time but rather the selection of minute subclones present at CLL diagnosis and years before overt transformation[21]. Additionally, recent studies focusing on DNA methylation (DNAm) further captured the genomic complexity of CLL[22–26], RS[27], de novo DLBCL[28–30], and other B cell neoplasms[31–33]. A better understanding of epigenetic signatures is needed, whether related to B cell development or tumor transformation mechanisms.

Distinguishing between CLL-derived RS and de novo DLBCL in a diagnostic setting based on histology and immunochemistry alone is challenging. Around 80% of RS cases are clonally related to the CLL disease stage while the remainder are unrelated (i.e. independent de novo DLBCL). This dichotomy is of importance for treatment decisions. De novo DLBCLs are chemosensitive in most patients, whereas CLL-derived RS is mainly characterized by chemoresistance and poor outcome, with a median overall survival (OS) of around 12 months.

In this study, we perform genome-wide DNAm analysis and whole-transcriptome profiling for a large series of primary human RS samples, and comprehensively compare our findings to those in CLL and DLBCL. We extensively characterize the epigenetic architecture of the RS samples and find the majority retain a CLL imprint. Remarkably, applying DNAm- and gene expression-based classifiers to datasets from landmark studies identifies a subset of "RS-type" DLBCL that is not previously described at the genomic level, is enriched in cases with an ABC-like COO signature, and has an unfavorable prognosis [7,8,10].

## Results
### Data quality controls
The study workflow is described in Fig. 1. We investigated DNAm using array-based technologies, exploring a total of 433 samples, including 58 RS samples, 25 CLLs paired with RS (i.e. tumor samples were available at both CLL and RS stages; hereafter "paired-CLLs"), 68 DLBCLs, and additional published methylomes from 190 other CLLs, and 92 samples representing normal B cell subpopulations (Supplementary Fig. 1)[22,25,26,34,35]. Limiting the batch effect is critical for comparing large cohorts explored with different platforms in different facilities. In this regard, we used EPIC and 450K Illumina microarray platforms, as these provide accurate, robust, and reproducible genome-wide coverage of CpG sites[36,37]. We extensively explored potential batch effects and showed it was completely removed after applying strict quality

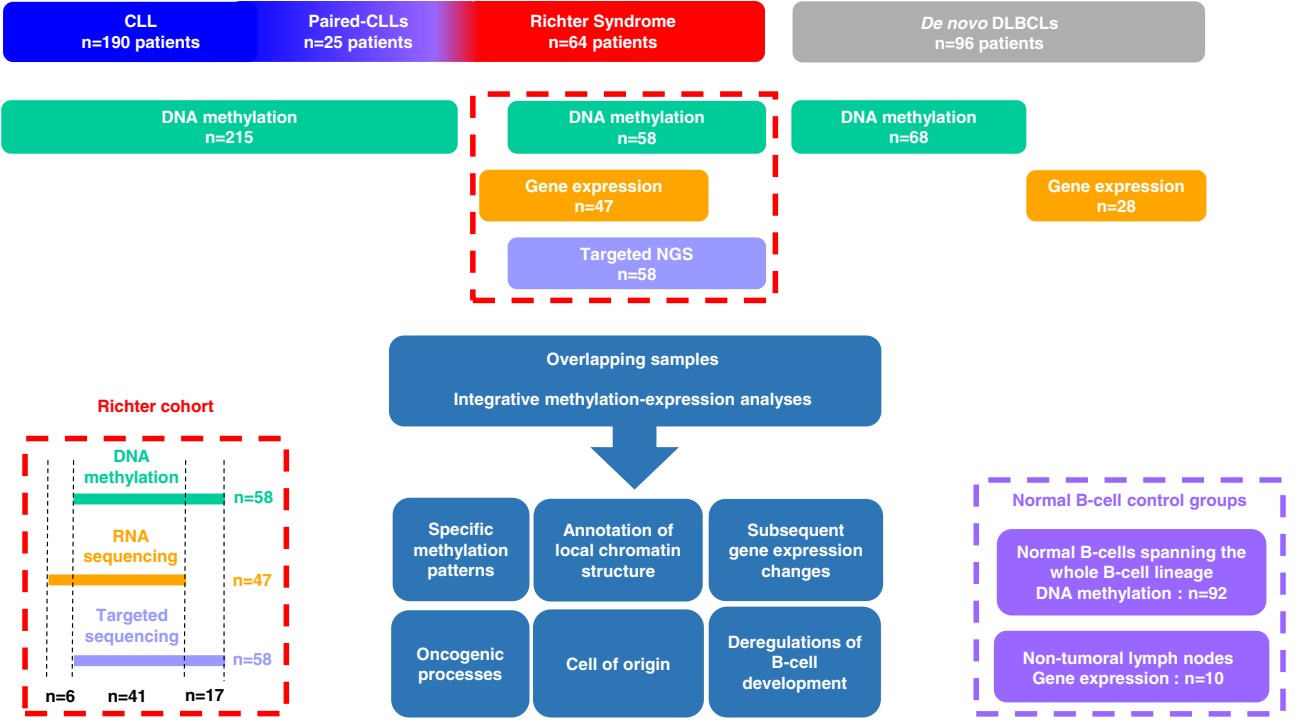

**Fig. 1 | Study workflow.** Genome-wide DNA methylation data were available for 58 RS, 25 CLLs paired with RS (tumor DNA samples were available at both CLL and RS stages), 190 other CLLs, 68 de novo DLBCLs, and 92 samples from normal B cells spanning the entire B lineage. All 58 RS samples were also documented for mutations in a custom panel of 13 CLL driver genes, and RNA-sequencing data were concomitantly available for 41 RS samples, allowing integrative analyses and

detailed exploration of oncogenic processes and epigenetic network deregulations. RNA sequencing data were obtained for another 6 RS, 28 de novo DLBCLs, and 10 non-tumoral lymph nodes. Data acquired from normal B cell control groups were used for methodologic purposes only (see "Methods"). CLL chronic lymphocytic leukemia, DLBCL de novo diffuse large B cell lymphoma, NGS next-generation sequencing, RS Richter syndrome.

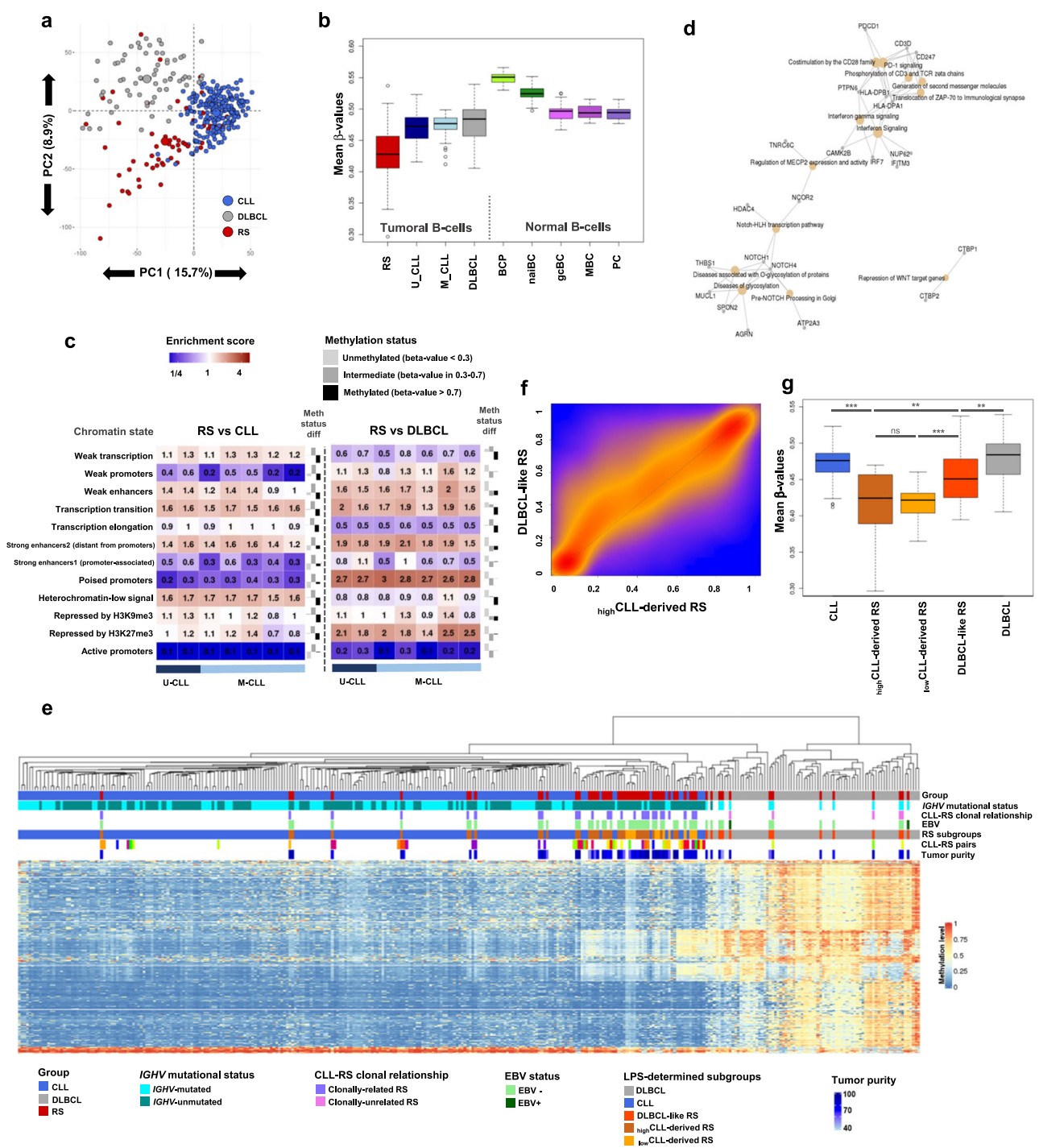

controls (see "Methods"). In addition, we applied a bioinformatic deconvolution method to separate methylation data attributable to five subtypes of normal white blood cells (CD4+ T-lymphocytes, CD8+ T-lymphocytes, neutrophils, monocytes, B cells). Use of respective cell composition data as covariates in supervised analyses limited the influence of tumor cell content of our samples.

## RS is a DNA hypomethylated entity versus CLL and de novo DLBCL

Unsupervised principal component analysis (PCA) showed a clear partitioning between RS, CLL, and DLBCL samples in the most variable components, highlighting different DNAm patterns in each group (Fig. 2a and Supplementary Figs. 2 and 3). Principal component

1 separated CLL from RS and DLBCL, while principal component 2 separated DLBCL from RS. However, some RS clustered within the DLBCLs or the CLLs. Decreased DNAm was observed in RS compared to CLL, DLBCL, and normal B cells (Fig. 2b and Supplementary Figs. 4 and 5). DNAm levels of the paired-CLLs were intermediate between RS and the other CLLs (Fig. S6). Hypomethylated and hypermethylated CpGs in RS were differentially distributed regarding CpG islands but similarly distributed regarding genomic context (Supplementary Figs. 7 and 8).

Next, we annotated CpGs differentially methylated between RS, CLL, and DLBCL according to 12 chromatin states reported in 7 CLL reference epigenomes[26]. The 102,614 CpGs differentially methylated between RS and CLL (two-way moderated *t* test adjusted for a false

**Fig. 2 | DNA methylation comparative analysis with CLL and de novo DLBCL shows that RS is a heterogeneous and hypomethylated entity. a** Unsupervised principal component analysis of the adjusted DNAm values of RS, CLL, and DLBCL. Geometrical centers are represented by bigger circles of the same color. **b** Boxplots of sample-averaged methylation levels with all 397,769 CpGs. RS ($n = 58$) versus U-CLL ($n = 112$): $p = 7.74e{-}11$; RS versus M-CLL ($n = 103$): $p = 4.46e{-}12$; RS versus DLBCL ($n = 68$): $p = 6.07e{-}12$. **c** Distribution of differential CpGs (FDR < 0.01; methylation differential >10%) according to the reported chromatin states in 7 CLL reference epigenomes[26]. Enrichments are shown as a heatmap and were calculated from the position of the selected CpGs. Their distribution was reported among 12 different chromatin state categories. Barplots in the right part of each panel show the methylation status difference in RS versus CLL or DLBCL. Differentially methylated CpGs are distributed among 3 methylation level categories. Upward bars indicate a comparative gain of CpGs in RS for the corresponding category, while downward bars indicate a comparative loss in RS. **d** RS versus CLL top annotations network (ReactomePA) from 238 differential DMRs computed with DMRcate (Fisher's comparison statistics: min_smoothed_FDR and HMFDR both <0.01; max beta-value differential >30%; at least 3 CpGs in the DMR with no gap >1 kb between CpGs). **e** DNAm-based linear predictor score (LPS) CpG architecture. Hierarchical clustering of 4863 CpGs differential between CLL and DLBCL (FDR < 0.01; beta-value differential >30%; moderated $t$ test). **f** Density map of DNAm between highCLL-derived and DLBCL-like RS. Smoothed beta-value densities from the EPIC dataset. Scale from blue (no density) to yellow (medium density) and red (high density). **g** Boxplots showing general methylation levels for highCLL-derived ($n = 33$), lowCLL-derived ($n = 12$), and DLBCL-like RS ($n = 13$), de novo DLBCLs ($n = 68$), and CLLs ($n = 215$). CLL versus highCLL-derived RS: $p = 2.2e{-}16$; highCLL-derived RS versus DLBCL-like RS: $p = 5e{-}3$; lowCLL-derived RS versus DLBCL-like RS: $p = 9.9e{-}3$; DLBCL-like RS versus DLBCL: $p = 3.5e{-}2$. BCP B cell precursors, CLL chronic lymphocytic leukemia, DLBCL de novo diffuse large B cell lymphoma, DNAm DNA methylation, EBV Epstein–Barr virus, FDR false discovery rate, gcBC germinal center B cells, highCLL-derived RS CLL-derived RS with a high LPS, HMFDR harmonic mean of the individual components FDR, MBC memory B cells, M-CLL IGHV-mutated CLL, lowCLL-derived RS CLL-derived RS with a LPS score below threshold, LPS linear predictor score, naiBC naive B cells, PC plasma cells, PC1/2 principal component 1/2, RS Richter syndrome, U-CLL IGHV-unmutated CLL. $p$ values were derived from two-sided $t$ tests. **$p < 0.01$; ***$p < 0.001$; ns not significant. For all box plots, center line indicates median; box limits indicate upper and lower quartiles; whiskers indicate 1.5× interquartile range; points indicate outliers. Source data are provided as a Source data file.

discovery rate (FDR) < 0.01; 90.8% hypomethylations in RS) were: depleted (ratio < 0.75) in active promoters, poised promoters, promoter-associated strong enhancers, and weak promoters; and enriched (ratio > 1.5) in transcription transition regions and heterochromatin (Fig. 2c). The 82,940 CpGs differentially methylated between RS and DLBCL (96.4% hypomethylations in RS) were: depleted in active promoters; and enriched in poised promoters and regions repressed by H3K27me3. Differentially methylated regions (DMRs; see "Methods") between RS and DLBCL were strongly enriched in targets of polycomb complex components SUZ12 ($p = 1.2e{-}121$) and EZH2 ($p = 1.5e{-}30$), which likely corresponds to the derivation of DLBCL from GC or post-GC B cells. Notably, genes associated with the extracellular matrix were overrepresented in this subset (Supplementary Fig. 9 and Supplementary Data 1). DMRs between RS and CLL were linked to NOTCH and Wnt pathways, and to the adaptive immune system, with PD-1 signaling and T cell/B cell co-stimulations (Fig. 2d and Supplementary Data 2), which likely corresponds to the driver role of NOTCH and PD-1 signaling in RS onset.

## DNA methylation separates CLL-derived and DLBCL-like RS subgroups

The PCA principal component 2 split the RS samples into two subgroups, one with a profile similar to CLL, the other closer to DLBCL (Fig. 2a). We postulated that "CLL-derived RS" (maintaining a CLL imprint) could be separated from "DLBCL-like RS" (distinct from the preceding CLL and closer to DLBCL). To test this, we modeled a linear predictor score (LPS)[38], computing two underlying probabilities (p): one to label samples according to their CLL-derived RS profile ($p_{CLL\text{-}derived}$), one for DLBCL-like RS ($p_{DLBCL\text{-}like}$), defining $p_{CLL\text{-}derived} \geq 98\%$ and $p_{DLBCL\text{-}like} \geq 98\%$ to obtain highly specific and homogeneous groups (see "Methods"; Supplementary Fig. 10). The statistical model devised to compute LPS was constructed with 4863 CpGs robust in separating CLL from DLBCL. Since de novo DLBCLs are usually IGHV-mutated whereas CLL may be IGHV-mutated or -unmutated, we excluded CpGs highly differential according to IGHV status[22,24] from the LPS calculation to focus on other distinctive features between CLL and DLBCL. The LPS scoring system was confirmed with hierarchical clustering (Fig. 2e), non-negative matrix factorization (NMF), PCA (Supplementary Figs. 11 and 12), and displayed differential patterns on normal cells spanning the B cell lineage (Supplementary Fig. 13). The scoring system identified 33 CLL-derived RS (57%) and 13 DLBCL-like RS (22%), leaving 12 intermediate samples (21%). This latter subgroup clustered within the CLL and CLL-derived RS branch, albeit marginally (Fig. 2e). The subgroup was then referred to as "low-LPS score CLL-derived RS" (lowCLL-derived RS), in contrast to the "high-LPS

score CLL-derived RS" (highCLL-derived RS). Comparing highCLL-derived RS and DLBCL-like RS confirmed global hypomethylation of highCLL-derived RS. In addition, DLBCL-like RS genomic distribution of DNAm did not coincide with that of DLBCL, with most locations hypomethylated in DLBCL-like RS (Fig. 2f, g and Supplementary Fig. 14). This subgrouping was not influenced by the tumor cell content (Supplementary Fig. 10).

## RS homogeneous subgrouping corroborates with gene expression

Among the 58 RS samples investigated for DNAm, 41 also underwent whole-transcriptome profiling. RNA samples from 6 independent RS cases were also sequenced. In total, the RNA-sequencing experiment included lymph node samples of 47 RS, 2 paired CLLs, and 28 DLBCLs, plus 10 non-tumoral samples for methodologic validation purposes (see "Methods"). Hierarchical clustering of the 23,508 identified genes confirmed clear subgrouping among RS samples (Fig. 3a). All RS classified as DLBCL-like RS by DNAm clustered with DLBCL (predominantly with non-GCB subtype) and separated from CLL-derived RS. This supports the existence of CLL-derived RS and DLBCL-like RS, through cross-validation using an orthogonal technique (>95% concordance). Annotations of gene clusters showed that CLL-derived RS shared a solid CLL gene expression signature, with upregulated genes involved in the BCR pathway and downregulated genes involved in the immune response, p53-signaling, and JAK-STAT pathways. Furthermore, $K$-means gene clustering of the 47 RS samples ranked according to LPS gradient revealed two main clusters of differentially expressed genes between highCLL-derived and DLBCL-like RS (Fig. 3b). One cluster is downregulated in highCLL-derived RS, is related to the extracellular matrix and TLR signaling, and included methylation-regulated p53 activity as an interesting feature (Supplementary Data 3). The other cluster is reminiscent of a CLL signature, overexpressed in highCLL-derived RS, and linked with NOTCH, PI3K signaling, and DNAm metabolism (Supplementary Data 4).

## RS subgroups correlate with IGHV mutational status and CLL-RS clonal relationship

To reduce the influence of IGHV mutational status on LPS, CpGs highly differential between U-CLL and M-CLL were filtered from the scoring CpGs. However, IGHV mutational status is associated with major DNAm changes in CLL[22,24,34]. Therefore, we next performed PCA on the 10,000 most variable CpGs, whether associated or not with GC reaction, tagging samples with IGHV annotations (Fig. 3c). CLL-derived RS accounted for nearly 80% of our RS samples and displayed a high

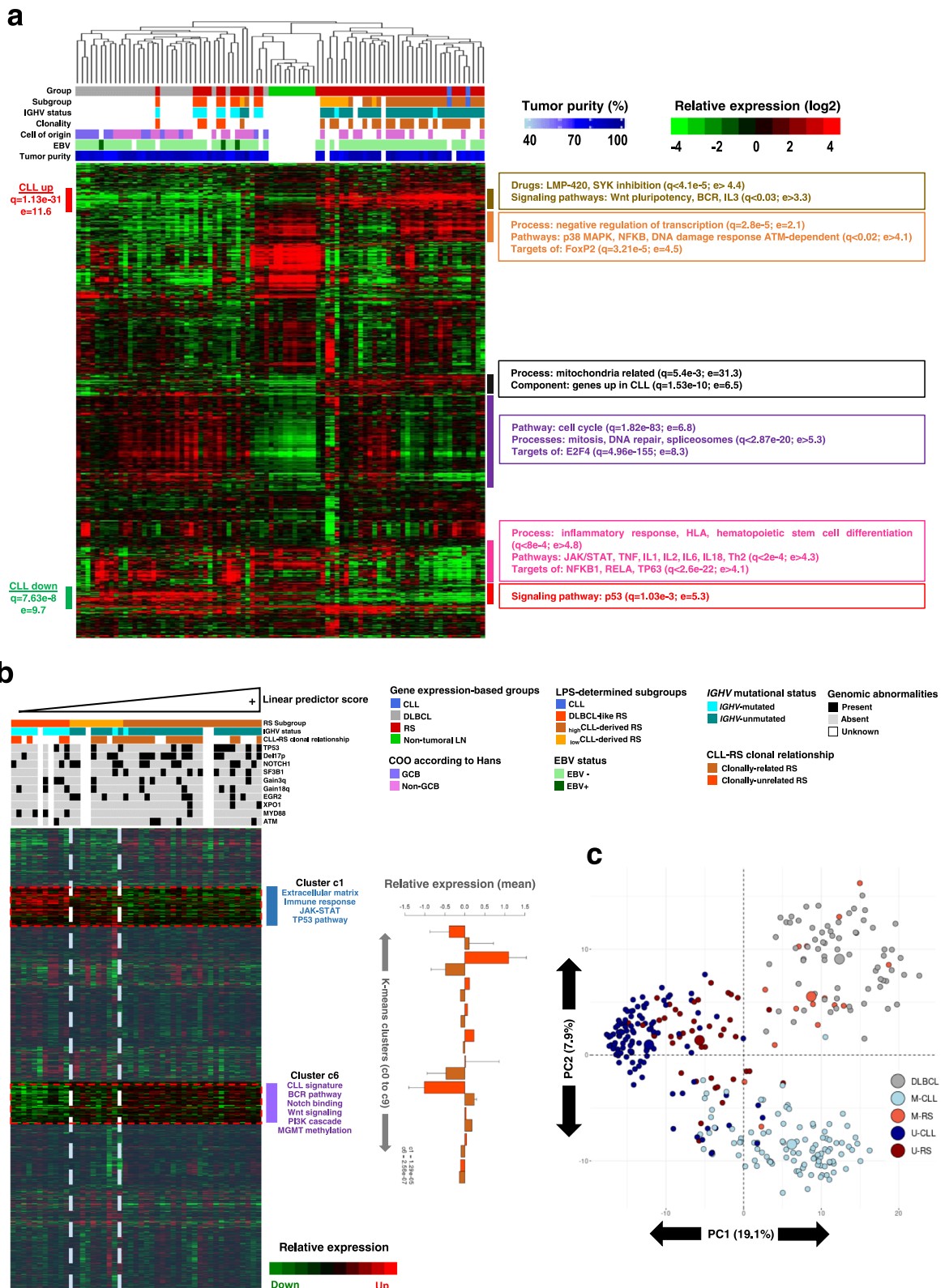

**a**

CLL up
q=1.13e-31
e=11.6

Drugs: LMP-420, SYK inhibition (q<4.1e-5; e> 4.4)
Signaling pathways: Wnt pluripotency, BCR, IL3 (q<0.03; e>3.3)

Process: negative regulation of transcription (q=2.8e-5; e=2.1)
Pathways: p38 MAPK, NFKB, DNA damage response ATM-dependent (q<0.02; e>4.1)
Targets of: FoxP2 (q=3.21e-5; e=4.5)

Process: mitochondria related (q=5.4e-3; e=31.3)
Component: genes up in CLL (q=1.53e-10; e=6.5)

Pathway: cell cycle (q=1.82e-83; e=6.8)
Processes: mitosis, DNA repair, spliceosomes (q<2.87e-20; e>5.3)
Targets of: E2F4 (q=4.96e-155; e=8.3)

Process: inflammatory response, HLA, hematopoietic stem cell differentiation (q<8e-4; e>4.8)
Pathways: JAK/STAT, TNF, IL1, IL2, IL6, IL18, Th2 (q<2e-4; e>4.3)
Targets of: NFKB1, RELA, TP63 (q<2.6e-22; e>4.1)

CLL down
q=7.63e-8
e=9.7

Signaling pathway: p53 (q=1.03e-3; e=5.3)

Tumor purity (%): 40  70  100

Relative expression (log2): -4  -2  0  2  4

Group / Subgroup / IGHV status / Clonality / Cell of origin / EBV / Tumor purity

**b**

Linear predictor score

RS Subgroup / IGHV status / CLL-RS clonal relationship / TP53 / Del17p / NOTCH1 / SF3B1 / Gain3q / Gain18q / EGR2 / XPO1 / MYD88 / ATM

Gene expression-based groups: CLL, DLBCL, RS, Non-tumoral LN

COO according to Hans: GCB, Non-GCB

LPS-determined subgroups: CLL, DLBCL-like RS, high CLL-derived RS, low CLL-derived RS

EBV status: EBV -, EBV+

IGHV mutational status: IGHV-mutated, IGHV-unmutated

Genomic abnormalities: Present, Absent, Unknown

CLL-RS clonal relationship: Clonally-related RS, Clonally-unrelated RS

Cluster c1
Extracellular matrix
Immune response
JAK-STAT
TP53 pathway

Cluster c6
CLL signature
BCR pathway
Notch binding
Wnt signaling
PI3K cascade
MGMT methylation

Relative expression (mean): -1.5 -1.0 -0.5 0.0 0.5 1.0 1.5

K-means clusters (c0 to c9)

c1 = 1.29e-05
c6 = 2.56e-07

Relative expression: Down — Up

**c**

PC2 (7.9%)

PC1 (19.1%)

DLBCL
M-CLL
M-RS
U-CLL
U-RS

prevalence of IGHV-unmutated samples. In contrast, 12/13 (93%) DLBCL-like RS were IGHV-mutated. RS subgrouping was thus highly associated with IGHV mutational status ($p = 6.3e−9$). This raises the possibility that RS subgroup partitioning simply reflects DNAm patterns of U-CLL and M-CLL. However, while most CLL-derived RS samples gathered among U-CLL, DLBCL-like RS samples regrouped with DLBCL, well separated from M-CLL (Fig. 3c).

Moreover, none of the DLBCL-like RS were clonally related to their respective CLL component ($n = 5$ pairs), confirming that DLBCL-like RS were not M-CLL-derived RS but rather de novo DLBCLs. In contrast, CLL epigenetic imprint is a feature of CLL-derived RS, likely an entity arising from CLL cells (Supplementary Fig. 15). This CLL-RS clonal relationship was further confirmed by identical IGHV-CDR3 sequences found in paired CLL and RS samples ($n = 26$ pairs; $p = 5.8e−6$). To

**Fig. 3 | RS gene expression profiles corroborate DNA methylation subgrouping.** **a** Unsupervised hierarchical clustering of RS and de novo DLBCL transcriptomes (RNA-Seq; 23,508 genes). **b** *K*-means consensus clustering of RS transcriptomes according to DNA methylation-based LPS gradient. Expression level statistics for each cluster are displayed as barplots. Barplot: data are presented as mean values +/− standard deviation from the mean. Cluster 1: $n = 1657$ genes; $p = 1.29e{-}5$. Cluster 6: $n = 2203$ genes; $p = 2.56e{-}7$. $p$ values were derived from two-sided $t$ tests. Source data are provided as a Source data file. Differential clusters are functionally annotated to the right. Mutational statuses as reported with NGS, or abnormalities determined with CNV analysis on DNAm data, are added below sample annotation for a selected panel frequently described in CLL and RS. **c** Sample partitioning according to IGHV mutational status. Unsupervised PCA clustering of U-RS, M-RS, U-CLL, M-CLL, and DLBCL according to the 10,000 most variable CpGs in the dataset. The focus is made on the most variable CpGs because these are highly representative of the IGHV signature in CLL (59% of these CpGs are strongly differential between U-CLL and M-CLL). Indeed, PC1 separates IGHV-unmutated from IGHV-mutated B cell malignancies, with U-CLLs and U-RS segregating in the same area. Conversely, M-RS partition with DLBCLs, clearly separated from M-CLLs on PC2. CLL chronic lymphocytic leukemia, COO cell of origin, DLBCL de novo diffuse large B cell lymphoma, DLBCL-like RS DLBCL-like Richter syndrome, e enrichment, EBV Epstein–Barr virus, GCB germinal center B cell, _high_CLL-derived RS CLL-derived RS with a high LPS, LN lymph node, _low_CLL-derived RS CLL-derived RS with an LPS score below threshold, LPS linear predictor score, M-CLL IGHV-mutated CLL, M-RS IGHV-mutated Richter syndrome, PC1/2 principal component 1/2, q $q$-value (corrected $p$ value), RS Richter syndrome, U-CLL IGHV-unmutated CLL, U-RS IGHV-unmutated Richter syndrome.

confirm the ability of the LPS to identify CLL-derived RS, we set up an independent validation EPIC 850 K experiment, investigating 52 samples (see "Methods" and Supplementary Fig. 16): (i) 44 new samples, including 18 new RS, the CLL component of 14 of these, 6 new DLBCLs, and 6 new CLLs; (ii) 8 samples from the first series: 4 RS samples (3 clonally related and 1 clonally unrelated), with the 4 respective CLL components. LPS classified 5/22 RS samples (22.7%; including the clonally unrelated RS from the first series) as DLBCL-like RS. Absence of clonal relationship with preceding CLL was confirmed by IGHV sequencing for 3 of these (data unavailable for the 2 other cases). The other 17 RS samples were identified as CLL-derived RS, with IGHV-assessed clonal relationship for 15/15 samples with concomitant CLL (Supplementary Fig. 16). These findings clearly indicate that DNAm is a powerful tool to determine the cellular origin in cases diagnosed as RS, as it differentiates DLBCL arising in a patient with CLL from true morphological transformations of CLL.

To further characterize our RS samples, we sequenced a panel of 13 CLL driver genes. Data integrated with copy number variations obtained from DNAm showed a high prevalence of CLL-driver mutations in RS samples harboring a CLL methylation signature (Supplementary Fig. 17). CLL-derived RS and DLBCL-like RS clinical features are displayed in Table 1. Both RS groups were uniformly treated with rituximab-based chemotherapy regimens, yet with inferior outcome for CLL-derived RS ($p = 1.7e{-}3$). This was further confirmed with gene-expression profiling, where RS samples aggregating in the CLL-derived branch of the dendrogram (Fig. 3a) were associated with a median OS of only 8 months. In contrast, RS samples clustering with the DLBCLs were associated with a longer median OS (35.5 months; $p = 0.018$) (Supplementary Fig. 18).

## CLL-derived and DLBCL-like RS feature different epigenetic networks

To better understand the epigenetic architecture of RS subgroups, we performed an integrative analysis based on correlations between DNAm and gene expression data (see "Methods"). The resulting integrome associated 674,567 transcripts with methylation loci. From these, 63,305 (9.4%) significant correlations ($p < 0.01$, Spearman's rho $<{-}0.33$ and $>0.33$) were first selected. Compared with DLBCL-like RS, _high_CLL-derived RS were mainly hypomethylated, which transcribed into a dominant direction of overexpression (Fig. 4a). Matching density maps were observed for _high_CLL-derived and _low_CLL-derived RS, with only slight differences. In contrast, DLBCL-like epigenomic programs largely differed (Supplementary Fig. 19), so we undertook an in-depth comparison of their integrome against that of _high_CLL-derived RS. Significant correlations between the two RS groups accumulated at regulatory locations and were mostly negative (77.3%; Fig. 4b). Genes under the control of these regions were related to cell proliferation (cell cycle, NOTCH pathway, PLCγ-mediated BCR signaling), epigenetic regulation and RNA processing, immune response (T- and B-lymphocyte activation and differentiation), and transcriptional regulation, including STAT family transcription factors (TF). Negative

correlations between promoter methylation levels and gene expression (rho $< {-}0.33$; at least three hits in the same regulatory region; Supplementary Data 5) led to a list of 666 unique associations showing enrichment in TF binding sites of SUZ12, TP63, TP53, and target genes of early B cell development TFs. Conversely, 234 regions correlated positively between DNAm and gene expression levels (22.7%; 3 hits with rho $>0.33$; Supplementary Data 6 and Fig. 4b). These were involved in controlling cellular proliferation and differentiation, regulation of transcription, protein metabolism, and immune response. Taken together, positively and negatively correlated locations amounted to 861 unique genes summarizing the most prominent features of _high_CLL-derived compared to DLBCL-like RS in terms of transcriptional mechanisms. Substantial differences in B cell development programs were highlighted, including the lower expression of B-lymphocyte-associated TFs *EBF1* and E2F partner *MSC/ABF1*, and the higher expression of *CD5*, *CCND1*, *ZAP70*, *ID3*, *BLK*, *WNT3*, *PRKCZ*, and *MGMT* in _high_CLL-derived RS (Fig. 4c, Supplementary Fig. 20, and Supplementary Data 7).

## Methylome and transcriptome integration provide insights into RS regulatory features
Key players of RS epigenetic deregulations were further identified in _high_CLL-derived RS, using DLBCL-like RS as a reference, and the 861 genes transcriptionally controlled through methylation. Among these, 156 were identified as TFs (18.1%; 2.3-fold enrichment; $p < 1e{-}16$)[39]. The regulatory network reconstructed in silico from these genes showed a central role of p53-like TFs and STAT proteins, an extensive control emanating from master regulators such as TP53, NF-KB1, and FOXC1, an essential developmental TF in many tissues which may have a role as a tumor suppressor. Over-represented target genes included those of the transcriptional repressors ZNF418 (6.1-fold; FDR = 1.87e{-}21) and ZNF217 (2.1-fold; FDR = 1.56e{-}8), involved in differentiation and antagonizing cell death, respectively. On the network, downstream effectors were mainly involved in epigenetic repression via the polycomb complex Prc2 (Supplementary Fig. 21 and Supplementary Data 8), for which we noted a SUZ12 signature (FDR = 5.68e{-}4) and an EZH2 target enrichment (FDR = 2.84e{-}4) in B cells, also linked with H3K9me3, H3K27me, and H3K27me3 epigenetic marks (FDR < 3.85e{-}6 in GM12878 cell line). The 156 TFs were strongly enriched in KRAB domain/C2H2-ZF-type TFs defining homeobox developmental proteins[40]. We observed P300 favored interactions (4.2-fold increase; FDR = 3.1e{-}3), denoting enhancers as enriched targets[41]. These results support our previous findings and highlight critical pathway reprogramming through selected epigenetic control of key TFs as an important mechanism in RS.

## RS-based classifiers uncover "RS-type" DLBCLs with poor outcome
DLBCL histological presentation of RS is essential to be distinguished from de novo DLBCL because they differ greatly in terms of prognosis. We thus developed a gene expression based linear classifier score

**Table 1 | Biological characteristics of the different RS subgroups, according to DNA methylation profiling**

| Characteristic | Full cohort | | CLL-derived RS | | DLBCL-like RS | | CLL-derived versus DLBCL-like RS |
|---|---|---|---|---|---|---|---|
| | n/N | % | n/N | % | n/N | % | |
| **Clinical features at CLL diagnosis** | | | | | | | |
| Age at diagnosis (years) | | | | | | | |
| Median (range) | 60 (35–82) | | 59 (35–80) | | 64 (52–82) | | p = 0.1 (NS) |
| Number of CLL treatment lines before RS transformation | | | | | | | |
| 0 | 18/56 | 32 | 10/44 | 23 | 8/12 | 66 | p = 0.02 |
| 1 | 14/56 | 25 | 12/44 | 27 | 2/12 | 17 | |
| ≥2 | 24/56 | 43 | 22/44 | 50 | 2/12 | 17 | |
| **Clinical and biologic features at RS diagnosis** | | | | | | | |
| Male (%) | 39/58 | 67 | 31/45 | 69 | 8/13 | 62 | p = 0.73 (NS) |
| Age at diagnosis (y) | | | | | | | |
| Median (range) | 66 (42–88) | | 65 (42–83) | | 69 (59–88) | | p = 0.12 (NS) |
| Time to RS transformation (y) | | | | | | | |
| Time <2 y | 15/56 | 27 | 10/44 | 23 | 5/12 | 42 | p = 0.44 (NS) |
| 2 y ≤ time ≤5 y | 10/56 | 18 | 8/44 | 18 | 2/12 | 16 | |
| Time >5 y | 31/56 | 55 | 26/44 | 59 | 5/12 | 42 | |
| **CLL status at RS diagnosis** | | | | | | | |
| Binet A | 34/50 | 68 | 27/40 | 68 | 7/10 | 70 | p = 0.41 (NS) |
| Binet B | 10/50 | 20 | 7/40 | 17 | 3/10 | 30 | |
| Binet C | 6/50 | 12 | 6/40 | 15 | 0/10 | 0 | |
| Response | 13/52 | 25 | 12/43 | 28 | 1/9 | 11 | p = 0.42 (NS) |
| Progression | 39/52 | 75 | 31/43 | 72 | 8/9 | 89 | |
| ECOG PS > 1 | 28/52 | 54 | 21/42 | 50 | 7/10 | 70 | p = 0.30 (NS) |
| Ann Arbor stage I–II | 8/55 | 15 | 7/43 | 16 | 1/12 | 8 | p = 0.67 (NS) |
| Ann Arbor stage III–IV | 47/55 | 85 | 36/43 | 84 | 11/12 | 92 | |
| **RS score** | | | | | | | |
| 0–1 | 30/49 | 61 | 21/39 | 54 | 9/10 | 90 | p = 0.07 (NS) |
| 2–3 | 19/49 | 39 | 18/39 | 46 | 1/10 | 10 | |
| **Rossi score[17]** | | | | | | | |
| High risk | 28/50 | 56 | 21/40 | 52 | 7/10 | 70 | p = 0.67 (NS) |
| Intermediate risk | 17/50 | 34 | 15/40 | 38 | 2/10 | 20 | |
| Low risk | 5/50 | 10 | 4/40 | 10 | 1/10 | 10 | |
| First-line RS treatment | | | | | | | |
| R-CHOP/R-ACVBP | 46/53 | 87 | 37/43 | 86 | 9/10 | 90 | p = 1 (NS) |
| Platinum-based immuno-chemotherapies | 7/53 | 13 | 6/43 | 14 | 1/10 | 10 | |
| **Response to RS first-line treatment** | | | | | | | |
| Complete remission | 15/53 | 28 | 10/42 | 24 | 5/11 | 45 | p = 0.35 (NS) |
| Partial remission | 2/53 | 4 | 2/42 | 5 | 0/11 | 0 | |
| Stable disease progression | 36/53 | 68 | 30/42 | 71 | 6/11 | 55 | |
| OS < 12 months | 42/56 | 75 | 35/44 | 80 | 7/12 | 58 | p = 1.7 × 10⁻³ |
| 12 ≤ OS ≤ 48 months | 8/56 | 14 | 8/44 | 18 | 0/12 | 0 | |
| OS > 48 months | 6/56 | 11 | 1/44 | 2 | 5/12 | 42 | |
| EBV positive | 3/21 | 14 | 1/16 | 6 | 2/5 | 40 | p = 0.12 (NS) |
| IGHV unmutated | 43/58 | 74 | 42/45 | 93 | 1/13 | 7 | p = 6.3 × 10⁻⁹ |
| Stereotyped IGHV | 12/58 | 21 | 10/45 | 22 | 2/13 | 15 | p = 0.71 (NS) |
| CLL clonally related | 26/31 | 84 | 26/26 | 100 | 0/5 | 0 | p = 5.8 × 10⁻⁶ |
| Large cell component (%), median [range] | 80 [50–95] | | 80 [50–95] | | 80 [50–90] | | p = 0.44 (NS) |
| Del 17p (13.1) | 26/58 | 45 | 23/45 | 51 | 3/13 | 23 | p = 0.11 (NS) |
| Del 11q (22.3) | 6/58 | 10 | 6/45 | 13 | 0/13 | 0 | p = 0.32 (NS) |
| Trisomy 12 | 11/58 | 19 | 9/45 | 20 | 2/13 | 15 | p = 1 (NS) |
| Del 13q (14.3) | 10/58 | 17 | 10/45 | 22 | 0/13 | 0 | p = 0.09 (NS) |
| TP53 | 21/58 | 36 | 17/45 | 38 | 4/13 | 31 | p = 0.75 (NS) |
| NOTCH1 | 21/58 | 36 | 18/45 | 40 | 3/13 | 23 | p = 0.33 (NS) |
| SF3B1 | 12/58 | 22 | 12/45 | 27 | 0/13 | 0 | p = 0.05 (NS) |
| EGR2 | 11/58 | 19 | 11/45 | 24 | 0/13 | 0 | p = 0.055 (NS) |

**Table 1 (continued) | Biological characteristics of the different RS subgroups, according to DNA methylation profiling**

| Characteristic | Full cohort | | CLL-derived RS | | DLBCL-like RS | | CLL-derived versus DLBCL-like RS |
|---|---|---|---|---|---|---|---|
| | n/N | % | n/N | % | n/N | % | |
| XPO1 | 7/58 | 12 | 7/45 | 16 | 0/13 | 0 | $p = 0.33$ (NS) |
| MYD88 | 5/58 | 8 | 1/45 | 2 | 4/13 | 31 | $p = 7 \times 10^{-3}$ |
| ATM | 4/58 | 7 | 4/45 | 11 | 0/13 | 0 | $p = 1$ (NS) |
| POT1 | 3/58 | 5 | 3/45 | 7 | 0/13 | 0 | $p = 0.1$ (NS) |
| RPS15 | 2/58 | 3.5 | 2/45 | 4 | 0/13 | 0 | $p = 1$ (NS) |
| FBXW7 | 1/58 | 2 | 0/45 | 0 | 1/13 | 8 | $p = 0.22$ (NS) |
| BIRC3 | 1/58 | 2 | 1/45 | 2 | 0/13 | 0 | $p = 0.4$ (NS) |
| BRAF | 1/58 | 2 | 1/45 | 2 | 0/13 | 0 | $p = 0.4$ (NS) |

Two-sided Student's t tests.

*CLL* chronic lymphocytic leukemia, *DLBCL* diffuse large B cell lymphoma, *EBV* Epstein–Barr virus, *ECOG PS* Eastern Cooperative Oncology Group performance status, *NS* non-significant, *OS* overall survival, *RS* Richter syndrome.

(LCS) to discriminate CLL-derived RS cases among DLBCL samples. We used a set of 215 genes selected from the transcriptomic CLL-derived RS signature (Supplementary Data 9; see "Methods") to screen external datasets of supposedly de novo DLBCL for the CLL-derived RS imprint. We first explored an independent gene expression dataset containing RS samples, untransformed CLLs, and EBV-positive DLBCL cell lines (GSE103265). The 215-gene set allowed unequivocal clustering of RS and CLL samples, well separated from DLBCLs (Supplementary Fig. 22). To cross-validate the previously described DNAm-based classifier (LPS) with the gene expression-derived classifier (LCS), we explored array-based DNAm and transcriptome-sequencing data of the ICGC MMML-Seq consortium (both classifiers can be used independently). Four (6.2%) DLBCL samples with classical DLBCL morphology showed extreme DNAm and gene expression scores, suggesting a CLL-like RS profile (Supplementary Fig. 23). Applying the gene expression-based classifier to array-based gene expression data of 430 DLBCL from the MMML-network identified 31 samples (7.2%) with a statistically significant score (see "Methods"). Next, we mined four large external cohorts of de novo DLBCL, including 1342 samples[8–10,42]. As with previous datasets, gene expression-based LCS distributions were biased toward overrepresenting extreme positive values (Supplementary Fig. 24). Our transcriptomic classifier identified 35/420 (8.3%; series from Lenz and colleagues)[42], 8/137 (5.8%; series from Chapuy and colleagues)[8], 13/223 (5.8%; series from Dubois and colleagues)[9], and 24/562 (4.3%; series from Wright and colleagues)[10] samples harboring the CLL-derived RS signature with a score above the threshold, for a total of 80/1342 (5.9%) samples. In the four datasets, 91.6% to 100% of these samples were of ABC-like subtype. We cross-compared this 215-gene signature and a discriminant 44-gene signature of ABC-type DLBCL[38], identifying *LMO2* as the only common gene. *LMO2* is an important gene of the ABC signature but holds no more weight in our classifier than the other 214 genes. Indeed, instead of just outlining every ABC-subtype DLBCL, our classifier extracted DLBCL with outstanding features, enriched in, but not exclusively, ABC-subtype DLBCL, with an overlap between ABC and GCB DLBCL and a subset of ABC-subtype DLBCL associated with a low LCS (Supplementary Fig. 25). DLBCL sharing the extreme score values with CLL-like RS showed a shorter progression-free survival (PFS) and/or OS ($p$ values ranging from $<10^{-3}$ to 0.02 depending on the cohort) compared to all other samples, and compared to other ABC-subtype DLBCL ($p$ values ranging from $<10^{-3}$ to 0.07) (Fig. 5a, b and Supplementary Figs. 26–28). We next conducted a multivariate analysis with Cox Proportional Hazards models, including all available covariates to evaluate the association of gene expression-based LCS with survival (OS and PFS). This association was set up in binary (top 25% versus the rest) as well as linear (as a continuous variable) models and provided estimates and effect size for each covariate (IPI, *TP53* and *MYC/BCL2* double hit status; Supplementary Fig. 29). This systematic analysis

confirmed strong associations with survival, independently from other covariates. In particular, this shorter survival was unrelated to international prognostic index distribution (Supplementary Fig. 30).

We next explored whether this effect might be due to the enrichment of a previously described genomic subgroup of ABC-like DLBCL associated with unfavorable prognosis[10]. In the 562-sample dataset from Wright and colleagues, the 25 cases with top LCS scores were enriched in formerly unassigned (1.5-fold relative enrichment; $p = 0.04$) and N1 subgroups (6-fold; $p = 6e-3$) while depleted in EZB subtype ($p = 0.03$). These cases were also strongly enriched (6.74-fold; $p = 4.1e-4$) in samples collected at relapse, raising the hypothesis of the ability of our classifier to identify DLBCL prone to relapse. Thus, the extreme LCS values seemed to characterize a distinct subset of ABC-type DLBCL, accounting for 4.3–8.3% of de novo DLBCL, with poor prognosis. The highest 25% scores in the series from Wright and colleagues showed biased distributions in genomic subgroups, dominated by unclassified cases, and associated with shorter PFS and OS (Fig. 6). These findings suggest an ability of the LCS classifier to: (i) identify high-scoring DLBCL samples as a separate DLBCL entity within de novo DLBCL, associated with ABC phenotypes and other features comparable to RS; and (ii) linearly classify other samples according to survival and overall prognosis (Supplementary Figs. 29 and 31). Interestingly, while absent from the 215-gene list, the CLL-associated marker *CD5* was overexpressed in RS versus DLBCL (2.4-fold; FDR = 2.13e−3) and high CLL-derived versus DLBCL-like RS (2.3-fold; FDR = 0.01). In the dataset from Wright and colleagues[10], *CD5* expression was higher in samples within the top 25% LCS than in other samples ($p = 5.8e-7$), corroborating our results. Last, in a dataset with concomitant transcriptome and CD5 immunochemistry staining GSE66770, the majority (17/22; 77.2%) of the top 25% samples were CD5+ DLBCL (2.1-fold enrichment) while this proportion was significantly lower (16/68; 23.5%) in the rest of the cohort ($p = 4.73e-3$).

## Discussion

In this study, by using genome-wide DNAm analysis and whole-transcriptome gene expression profiling, we extensively characterized the epigenetic architecture of primary human RS samples. We identified a CLL epigenetic imprint that can act as a surrogate for identifying whether an RS is clonally related to CLL or has arisen de novo. Discovery of the CLL imprint in an RS sample avoids reliance on obtaining tumor DNA at the CLL stage. Considering de novo DLBCL, DNAm- and gene expression-based classifiers delineated an RS-like subset in datasets from several landmark studies that was not previously described at the genomic level, was enriched in cases with an ABC-like COO signature, and had an unfavorable prognosis[7,8,10].

Previous extensive explorations with exome or full genome-sequencing had found differences in genomic landscapes between DLBCL-subtype RS and de novo DLBCL[16–20]. Here we used a different

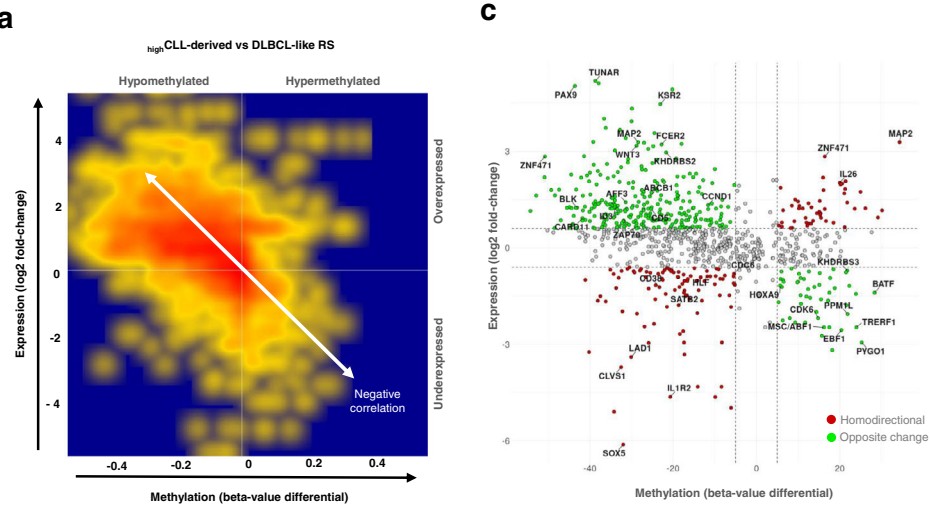

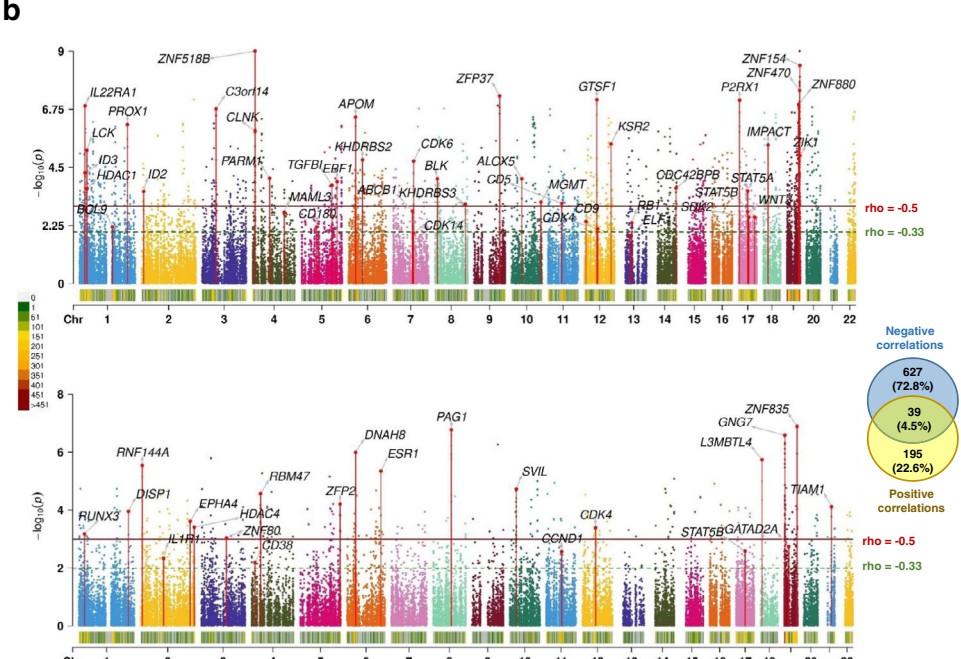

**Fig. 4 | Integrative analysis of DNA methylation and transcriptome data highlights different epigenetic programs in _high_CLL-derived and DLBCL-like RS.** **a** Density map (smoothed density scatterplot) representing overall DNA methylation versus gene expression changes between _high_CLL-derived RS and DLBCL-like RS. Scale ranges from blue (no density), to yellow (medium density) and red (high density). Only genes with at least one significant correlation (Spearman's test; _p_ value <0.01) were retained. Locations of the corresponding CpGs were mainly distributed in proximal and distal regulatory regions, with specific enrichments in TSS features for negative (TSS200: 2.6-fold, TSS1500: 2.2-fold) and positive (TSS200: 1.2-fold, TSS1500: 1.6-fold) correlations. Hypo/hyper-methylations and under/over-expressions are indicated relatively to the _high_CLL-derived RS subgroup. **b** Manhattan plots of negatively and positively correlated regulatory regions and associated transcript expressions. Chromosomes are displayed at the bottom of each plot, with a color code (from green to red) indicating the density of

correlations over sliding windows of 1 Mb. Series of vertically aligned dots indicate DMRs (of at least 3 CpGs with a hit in TSS-associated location) significantly correlated with gene expression. Upper part: negative correlations, amounting to 666 unique genes; bottom part: positive correlations, amounting to 234 unique genes; a VENN diagram indicates the overlap between negative and positive correlations. **c** Quadrant scatterplot displaying methylation levels of regulatory sequences and corresponding expression levels for the 861 selected genes (overall absolute correlations: rho = 0.72; _p_ < 2.2e−16; Spearman's tests). The upper left and lower right quadrants show genes with a negative correlation between methylation and expression. Lower left and upper right areas: genes with positive correlations. CLL chronic lymphocytic leukemia, DLBCL de novo diffuse large B cell lymphoma, DLBCL-like RS DLBCL-like Richter syndrome, DMR differentially methylated region, _high_CLL-derived RS CLL-derived RS with a high linear predictor score, RS Richter syndrome, TSS transcription start site.

---

study design and methodological approach to expand this knowledge. Firstly, we studied epigenetic deregulations using robust and proven methods, and profiled the RS molecular landscape beyond gene mutations and copy number variations. Secondly, we conducted a comprehensive analysis of RS pathophysiology which combined the analysis of genome-wide DNAm and whole transcriptome profiling,

rather than pinpointing a limited number of specific targets. Thirdly, we compared the RS epigenetic profile to that of large cohorts of diverse CLL and de novo DLBCL, which contrasts with previous work mostly focusing on the RS transformation process.

Human-derived xenograft mouse models and cell lines were recently reported to study RS biology and test drug response[11–15].

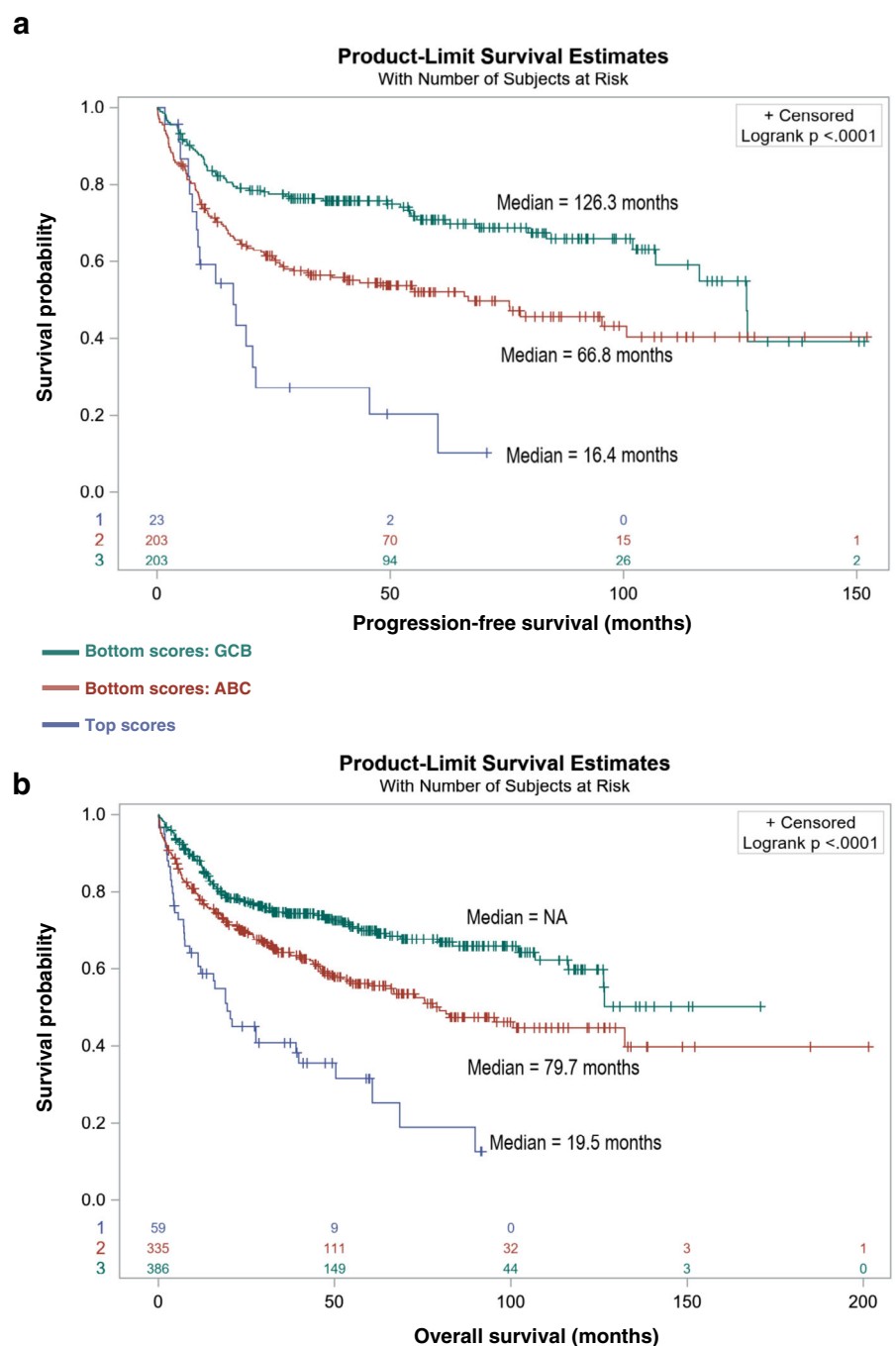

**Fig. 5 | DLBCLs harboring the CLL-derived RS epigenetic signature are associated with ABC phenotype and worse outcome. a** Kaplan–Meier estimates of progression-free survival for $n = 429$ patients from three combined and clinically annotated public DLBCL datasets[8-10]. Comparative PFS between patients with top LCS and the rest of the cohorts, according to COO ($p = 8.4e-8$). **b** Kaplan–Meier estimates of overall survival for $n = 780$ patients from four combined and clinically annotated DLBCL public datasets[8-10,42]. Comparative OS between patients with top LCS and the rest of the cohorts, according to COO ($p = 1.1e-11$). Statistical comparisons were performed with the log-rank test. Bonferroni method was used for multitesting adjustments. Datasets: from Lenz et al. ($n = 420$; microarray, accession under GSE10846; PMID: 21546504); from Chapuy et al. ($n = 137$; microarray, accession under GSE98588; PMID: 29713087); from Dubois et al. ($n = 223$; microarray, accession under GSE87371; PMID: 31648986); from Wright et al. ($n = 562$; RNA-Seq; PMID: 32289277). ABC activated B cell, CLL chronic lymphocytic leukemia, COO cell of origin, DLBCL de novo diffuse large B cell lymphoma, GCB germinal center B cell, LCS linear classifier score, OS overall survival, PFS progression-free survival, RS Richter syndrome.

However, the availability of these models is limited and they cannot recapitulate the full heterogeneity of RS, as they were generated from a limited number of tumor samples. Our approach using large cohorts of primary human RS samples and comparative tumor material also holds promise for discoveries and better characterize the wide RS epigenetic complexity. We cross-validated our epigenetic findings using DNAm patterns, that were largely corroborated by transcriptome data, in an independent manner.

Our genome-wide DNAm data provide a more complete RS hypomethylation profile description. The DNAm patterns confirm previous findings that RS is a DNA-hypomethylated entity as compared with CLL and de novo DLBCL[27]. Such global hypomethylation may in

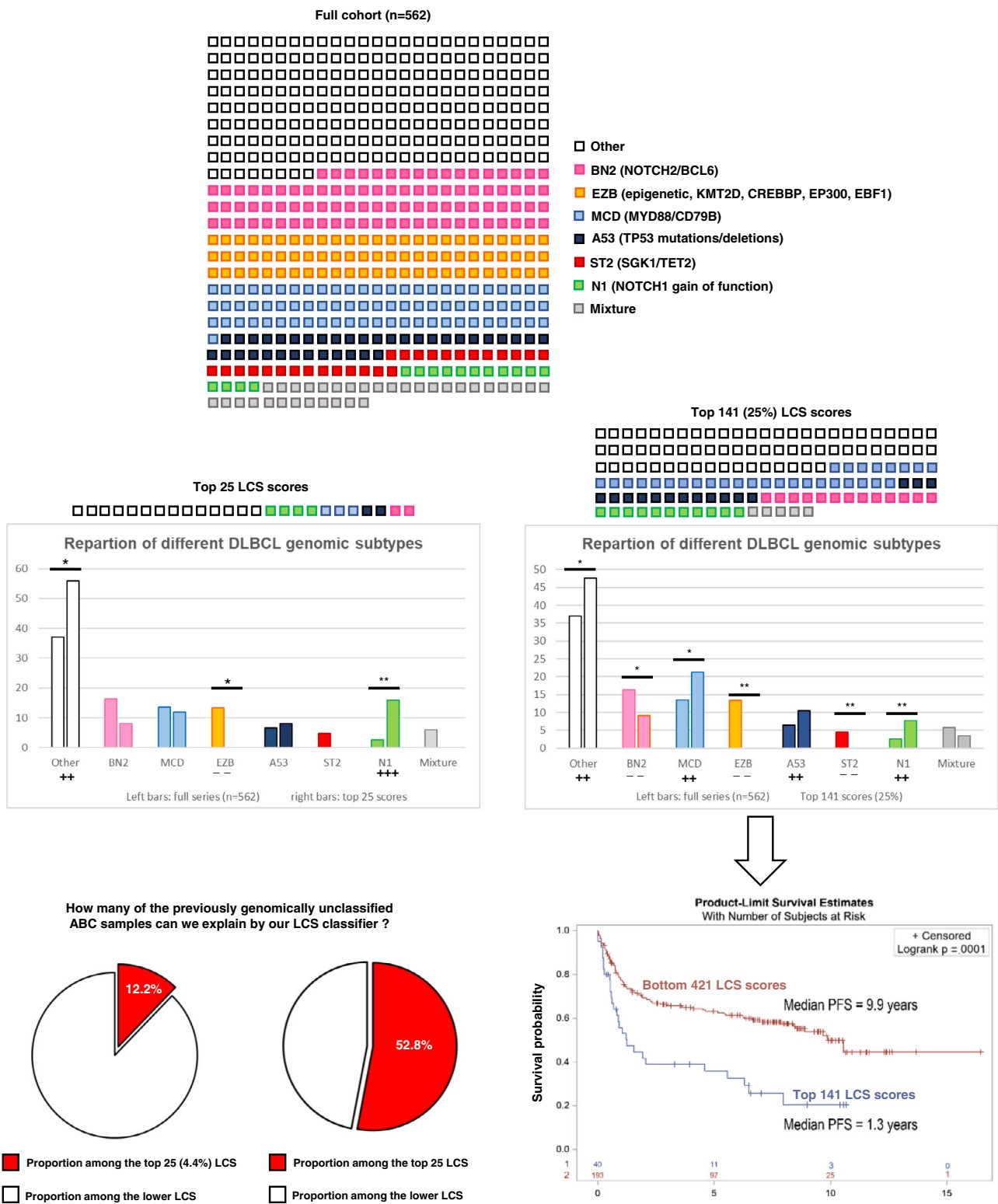

part reflect a more extensive proliferative history of the RS subclone[21], as measured by the epiCMIT mitotic clock[33]. Using a reproducible DNAm microarray uniformly spanning the vast majority of regulatory regions at a whole-genome scale[37], we characterized the epigenetic architecture underlying the commonly accepted dichotomic heterogeneity with regard to whether a primary RS is clonally related to CLL or has arisen de novo[17]. As expected, around 80% of our RS samples harbored a CLL epigenetic imprint (likely derived from a pre-existing CLL clone). This was confirmed by identical IGHV-CDR3 sequences for all

CLL-RS follow-ups. As nearly all de novo DLBCLs harbor a mutated IGHV, we propose that RS clonally related to the underlying CLL clone are: (i) IGHV-unmutated DLBCL; and (ii) IGHV-mutated DLBCL with a CLL imprint. Determining CLL history using DNAm and gene expression by identifying a CLL imprint independently from matched-CLL availability is a step forward, and is essential for clinical and therapeutic management. Interestingly, DLBCL-like RS would conversely be DLBCL without clonal relationship with the CLL counterpart. However, DNAm of DLBCL-like RS differed from that of de novo DLBCL in terms of

**Fig. 6 | The gene expression-based LCS linearly classifies de novo DLBCL samples, with high scores enriched in N1, unclassified genomic profiles[10], and shorter progression-free survival.** Dataset from Wright et al. (*n* = 562; RNA-Seq; PMID: 32289277). Two-sided *t* tests were used to assess statistical significance. Top 25 LCS scores: enrichment in "other" subtype (*e* = 1.51; *p* = 4.6e−2); depletion in EZB subtype (*e* = 0; *p* = 3.0e−2); enrichment in N1 subtype (*e* = 5.99; *p* = 6.4e−3). Top 141 (25%) LCS scores: enrichment in "other" subtype (*e* = 1.28; *p* = 1.4e−2); depletion in BN2 subtype (*e* = 0.56; *p* = 1.9e−2); enrichment in MCD subtype (*e* = 1.57; *p* = 1.7e−2); depletion in EZB subtype (*e* = 0; *p* = 1.7e−8); enrichment in A53 subtype (*e* = 1.62; *p* = 7.5e−2); depletion in ST2 subtype (*e* = 0; *p* = 2.6e−3); enrichment in N1 subtype (*e* = 2.92; *p* = 7.0e−3). Survival curves: Kaplan-Meier estimates of progression-free survival for *n* = 233 patients from a clinically and genomically annotated dataset from Wright and colleagues. Comparative PFS between patients with top 25% LCS and the rest of the cohort. Statistical comparisons were performed with the log-rank test (*p* = 1e−4). Source data are provided as a Source Data file. ABC activated B cell like, A53 *TP53* mutations/deletions-associated DLBCL subgroup, BN2 DLBCL subgroup associated with lesions of *BCL6* and/or *NOTCH2*, COO cell of origin, DLBCL de novo diffuse large B cell lymphoma, EZB DLBCL subgroup associated with abnormalities of epigenetic regulators *KMT2D*, *CREBBP*, *EP3OO*, and/or *EBF1*, GCB germinal center B cell, LCS linear classifier score, MCD DLBCL subgroup associated with lesions of *MYD88* and/or *CD79B*, N1 DLBCL subgroup associated with *NOTCH1* gain of function, PFS progression-free survival, RS Richter syndrome, ST2 DLBCL subgroup associated with lesions of *SGK1* and/or *TET2*. *\*p* value <0.05; *\*\*p* value <0.01; ++: enrichment >1.2; +++: enrichment >5; −: depletion <0.6.

increased cell cycle activity and IGF1, ERK/MAPK, PI3K/AKT, and PD-1 signaling pathways. These differences suggest influences of the CLL-invaded microenvironment for the development of a specific DLBCL pathogenesis[43].

Moreover, by integrating the DNAm and transcriptomic data, we evidenced different epigenetic networks in CLL-derived and DLBCL-like RS. Epigenetic architecture remodeling and subsequent deregulation of EZH2 and Wnt pathways, as well as PI3kinase/AKT and IGFR1 signaling cascades, unravel CLL-derived RS underlying mechanisms potentially responsible for chemotherapy resistance. These mechanisms are potentially druggable through EZH2, PI3K/AKT, or IGFR1 inhibitors. IGFR1 pathway triggering was recently described as a resistance mechanism to targeted therapy in CLL[44]. Interestingly, O6-methylguanine-DNA methyltransferase *MGMT* regulatory sequences are hypomethylated and *MGMT* is consequently overexpressed in CLL-derived RS. *MGMT* promoter hypomethylation status is a known negative prognostic marker in glioblastoma[45], de novo DLBCL[46], and an actionable target. This marker is easily assessable in the context of DLBCL diagnosis and routinely used to guide therapeutic decisions.

Our results show B cell-specific TF implication and epigenetic imprint in CLL-derived RS, and emphasize the previously described important role of TP53, FOXC1, NF-KB, and epigenetic regulators in oncogenic mechanisms. Strikingly, genes involved in the regulation of TP53 activity through methylation were overexpressed in CLL-derived RS, confirming the central role of TP53 in clonally-related RS and the primary importance of epigenetic deregulation in the transformation process. An interesting finding of this study is the putative role of the FOXC1 TF in the RS regulatory network. FOXC1 has previously been described as cooperating with HOX family members for orchestrating mesenchymal tissue development, through NF-KB signaling[47]. FOXC1 is PRC2 repressed during hematopoietic development, but frequently derepressed in hematopoietic progenitors in acute myeloid leukemia[48]. Our data identified FOXC1 derepression as a hallmark of CLL-derived RS, likely associated with the blockade of B cell development and proliferation due to NF-KB signaling unleashing. We also observed hypomethylation of DMRs regulating the expression of genes involved in the extracellular matrix organization, and in the immune system. These observations suggest a strong influence of the microenvironment in RS development.

Notably, our findings directly translate into classification and prognostication of de novo DLBCL, the most common human B cell lymphoma. We provide a gene expression-based, stable, reproducible, and potentially widely applicable classifier, on the basis of a CLL-derived RS epigenetic imprint. The classifier differentiates a particular DLBCL subgroup from supposedly de novo DLBCL datasets. Of clinical importance, cases assigned to this subgroup are frequently not detected by recently described genomic and gene expression classifiers of DLBCL, and they are associated with an unfavorable prognosis. These cases were ABC-like DLBCL, enriched in unclassified or N1 DLBCL genomic subtypes[7,10]. This is in line with the association of RS with a particular gene expression profile and with *NOTCH1* mutations and NOTCH pathway activation. Given the efficacy of ibrutinib plus R-CHOP chemotherapy in N1 subtype DLBCL[49], the enrichment in N1 profile within RS samples supports research into whether these patients may also benefit from BTK inhibition combined with R-CHOP chemotherapy. However, a recent single cell transcriptome analysis of sequential CLL-RS samples revealed that, as compared to the CLL cells, RS cells downregulate genes related to BCR signaling and upregulate those involved in oxidative phosphorylation[21], and therefore RS may be less sensitive to Ibrutinib. By applying a stringent cut-off to our transcriptomic score, generalized to all studied DLBCL datasets[8–10,42], we identified a separate de novo DLBCL subset associated with a median PFS comparable to that of clonally-related RS. Based on our observations, 4-8% of DLBCL diagnosed as de novo DLBCL, non-otherwise specified, may in fact be a subgroup of DLBCLs sharing common epigenetic and transcriptional features with clonally related RS, and with a similar unfavorable outcome. We propose a stable and reproducible expression-based classifier widely applicable to transcriptomic data, enabling the identification of this specific entity within supposedly de novo DLBCL, termed "RS-type DLBCL." Limitations of the transcriptome scoring method are dataset size and composition (DLBCL features associated with outcome), which by design prevent the exploration of single samples independently and may exert biases. However, the method also demonstrated the linear association of DLBCL scores with poor outcome and clinical variables of cancer aggressiveness, and so constitutes a means for improving the current DLBCL classification system.

In conclusion, our study has revealed several relevant aspects of RS biology, including the complete RS hypomethylation profile and differentiation of clonal versus non-clonal RS according to DNAm patterns and gene expression profiles. The discovery of a CLL imprint allows clonal relationship assessment without the need for tumor DNA at the CLL stage. Subgrouping of primary RS samples according to extensive characterization of the epigenetic architecture has provided information underlying oncogenic processes, with clear clinical implications. In particular, identification of RS-type DLBCL cases helps to advance the current DLBCL classification system and could be incorporated in treatment decisions, potentially improving disease management. Our findings also enable the evaluation of larger cohorts recruited in clinical trials and the development of novel treatment approaches, which are urgently needed in RS.

## Methods
Our methods and results made extensive use of data from previous landmark studies[26,31]. Care was taken to follow good practices in the analyses of methylome and transcriptome data, employing widely approved procedures previously used in other high-standard studies. Regarding the handling of large cohorts, we used sample correlations, performed genotype checks between omics data, and added technical and biologic replicates wherever possible.

### Ethics statement
This study complies with all relevant ethical regulations and we have obtained written informed consent for all participants. No

compensation was provided. We obtained consent to use and publish information that identifies individuals, including indirect identifiers such as gender and age. Individuals recruited for this study can no longer be identified by the information provided, due to sample anonymization and processing of the genomic data. All procedures were in accordance with Helsinki declaration. Study protocol was approved by the Institutional Review Boards and Ethics Committees of Nancy, Kiel (#A150/10), Ulm (#349/11; #459/19 and #96/08) and Barcelona university hospitals, and by the French national ethics committee (Comité de Protection des Personnes Ouest IV 09/05/2017).

## Patients and materials

A multicenter registry of RS accrual was established across nine centers affiliated to the French Innovative Leukemia Organization (ClinicalTrials.gov Identifier: NCT03619512). Sixty-four patients diagnosed with DLBCL-subtype RS were enrolled. Fresh frozen biopsies were gathered at RS diagnosis and met the criteria for DLBCL, including diffuse patterns of large B cells with the same size as macrophages or twice the size of normal lymphocytes[3,50]. For all patients, diagnoses were reviewed and confirmed by two independent pathologists. Only RS samples with at least 50% (median 80%, range 50-95%) high-grade component assessed by pathology review were selected for analysis. The same process was applied for assembling a validation cohort of 58 samples, further reduced to 52 QC-passed samples, which we processed to an independent EPIC 850K experiment. This 52-sample validation cohort included 44 new samples: 18 new RS samples, the CLL component of 14 of these, 6 new DLBCL samples, and 6 additional CLLs. In addition, 8 samples from the training series were used as controls: 4 RS samples (3 clonally related and 1 clonally unrelated), with the 4 respective CLL components (Supplementary Fig. 16). Thus, this EPIC 850K experiment investigated 22 RS, 6 new DLBCLs, and 24 CLLs, including 18 paired-CLLs.

Fifty-eight of the 64 enrolled patients with RS were from a previously described cohort, and both targeted NGS sequencing and DNAm exploration were performed; 56 of these 58 patients with RS underwent 18F-fluorodeoxyglucose positron emission tomography/computed tomography for initial diagnosis[51]. For the other six patients with RS, the fresh frozen biopsy was too small for extracting both DNA and RNA, and due to the large cellular component (>70%), we prioritized gene expression data and only RNA sequencing was performed. The minimal tumor purity was raised to 70% for RNA analysis, as contamination by signal from residual normal cells strongly influences global gene expression, especially for a subset of transcripts with very low expression in tumor cells but high expression in residual normal cells.

Additional data for CLL ($n = 215$), and 92 normal B cells spanning the entire B lineage development were obtained as part of previously published studies[22,25,26,34,35]. DNA methylation from 68 de novo DLBCL cases were also used as a reference. These DLBCLs originate from a larger lymphoma cohort gathered by the ICGC MMML-seq consortium[52]. Finally, 10 lymph nodes from healthy subjects were analyzed as a control group for transcriptome sequencing.

## Methylome data analyses

**EPIC microarray.** DNAm status of 866,562 CpG sites was interrogated on the *Infinium Methylation EPIC array* (Illumina, San Diego, CA, USA; see Supplementals), later referred to as the EPIC 850K platform.

**Dataset generation.** Datasets were created using the minfi package[53]. The EPIC set comprises 90 distinct samples (58 RS, 25 CLL, plus a subset of 7 DLBCL replicates also available on 450K), interrogated on EPIC 850K. DNA methylation data from the control groups (215 CLL, 68 DLBCL, 92 normal B cells spanning the entire B-lineage) were acquired with the Illumina Infinium® HumanMethylation450 BeadChip (later referred to as the 450K platform)[22,25,26,34]. These and the EPIC 850K data were processed from IDAT files. Analyzes were run under R 3.6 with Bioconductor 3.10 and later versions. The FULL dataset comprises 433

distinct samples (92 benign B cells, 215 CLLs, 68 DLBCLs and 58 RS), combined into a single 450K object containing probes shared by 850K and 450K microarrays: (i) raw IDAT files corresponding to 96 and 377 samples for the 850K (866,091 CpGs) and 450K (485,512 CpGs) platforms, respectively, and included technical replicates; (ii) each subset was loaded independently, stored into a dedicated *RGChannelSet* minfi object, along with full sample annotations, then both were combined into a third subset containing 473 samples × 452,567 CpGs using the *combineArrays* function with output type as "IlluminaHumanMethylation450k"; (iii) the EPIC dataset stems from the first (850K) subset alone, the FULL dataset is obtained from the combined subsets.

To reduce technological issues and biases, the same preparation protocol was applied to both EPIC (850K) and FULL (combined) subsets. The main stages of the filtering and quality control pipeline are as follows: (i) technical checks, filtering, and evaluations (ii) data normalization with SWAN[54]; (iii) probes located on X and Y chromosomes, flagged as cross-hybridization probes, or located near known SNPs were further removed with the *rmSNPandCH* function (with parameters dist = 2 and mafcut = 0.05) available from the DMRcate package[55]; (iv) imputation of the remaining failed $\beta$-value positions with *imputePCA* of the R missMDA package[56,57], (v) 2 × 2 sample correlation checks (Supplementary Figs. 32 and 33). Correlation heat maps were rendered with the R corrplot package; (vi) extended quality control step to remove sample outliers and check for residual postnormalization batch effects (Supplementary Fig. 34); (vii) ultimately, technical replicates were averaged into unique samples as all replicates were found comparable (Supplementary Fig. 35). These filtering steps led to the final EPIC (90 samples × 794,927 CpGs) and FULL (433 samples × 397,769 CpGs) datasets.

**Technical checks, filtering, evaluations, and quality control.** These steps included failed CpGs removal (>10% samples with a detection $p$ value >0.01), gender check between clinical data and gender returned by the *getSex* function, and genotype checks (Supplementary Data 10) between RNA-seq data (see Supplementals) and genotypes inferred with the *beta2genotype* function available from the R OmicsPrint package[58].

**Cell composition deconvolution.** Cell type composition was estimated for each sample with the *estimateCellCounts* function against a library of 6 normal white blood cells (CD8 T cells, CD4 T cells, NK, B cells, monocytes, and granulocytes) (Supplementary Data 11). The proportions of each explored cell type were reported and later used as covariates in statistical models to adjust for B cell representation in the mixes. Blood samples deprived in B cells (<30%) were thus discarded from further analyses.

## Downstream bioinformatics

**Supervised analyses.** As a rule, $\beta$-values were used for direct interpretation and graphical representation, while $M$-values were favored for statistics and computations. Linear modeling based on empirical Bayesian methods was used to assess for CpG differential methylation. When applicable, these models included cell deconvolution results as added covariates to correct for B cell content. Additionally, at this point, any unwanted methylation variation such as residual batch effects were removed by using the *RUVm* function from package missMethyl[59]. The overall dispersion was calculated on the entire dataset, then $p$ values for each comparison were obtained with a two-way moderated $t$ test and adjusted for FDR following the Benjamini−Hochberg procedure. At probe level, an FDR < 0.01 indicated statistical significance. Differentially methylated region (DMR) determination was performed on the same linear models with *dmrcate* (package DMRcate), with lambda = 1000 and C = 3. FDR cut-off for first allowing a CpG to initiate a DMR was set to FDR = 0.01, and DMRs were

considered statistically significant if both min_smoothed_fdr and HMFDR output probabilities were <0.01.

**Unsupervised analyses.** Explorations were conducted on $\beta$-values, and all methods used Euclidean distances as (dis)similarity metrics. PCAs were performed with R packages FactoMineR and factoextra, on the entire datasets or a subset of the top variant CpGs across all considered samples. Hierarchical clusterings included complete and average linkage criteria, and resulting heat maps and dendrograms were rendered with the R package ComplexHeatmap. Non-Negative Matrix Factorizations were performed with the R package NMF, either on all CpGs or a subset of the most variant ones according to the context, with method *lee*, and parameters ranking = 3 and iterations = 50.

**Feature annotations.** Methylome data were analyzed using the available Illumina 450K and EPIC platform annotations, which strongly rely upon the hg19 assembly. As several tools like minfi and DMRcate still use those by default, CpG and DMR locations/annotations were lifted to hg38 coordinates as an after-computation-process when required, especially when dealing with integrations with transcriptomic and epigenomic data. Additional CpG annotations included B cell development modules[34], UCSC tracks for the EBV-transformed GM12878 cell line, such as DNaseI and chromatin marks from ENCODE and transcription factor ChIP-Seq peaks from ENCODE3, histone modifications, and chromHMM chromatin states for 7 reference CLL epigenomes (2 U-CLLs and 5 M-CLLs)[26]. Any *liftOver* of coordinates between hg19- and hg38-annotated data was achieved with the UCSC table browser or with R packages liftOver and XGR.

**Gene set and pathway analyses.** Unbiased functional annotations on ontological terms (GO) and KEGG pathways were achieved at CpG and DMR levels with the R package missMethyl. Additional enrichment analyses were conducted on curated gene lists with Enrichr[60]. Reactome pathway overrepresentations and enrichment analyses were performed with ReactomePA[61] on curated sets of unique genes associated with identified DMRs. Enrichments in sets of CpGs, DMRs, target genes, GO terms, or pathways were calculated as the occurrences of the selection against a background representing the entire dataset (enrichment = observed frequency/expected frequency). *p* values associated with enrichment analyses were obtained with (i) an over-representation test, (ii) Fisher's exact test, or (iii) a Chi-square test, depending on the context and group size.

**Linear predictor score (LPS).** To formally distribute RS samples into subgroups, we developed a scoring predictor inspired by the work of Wright and colleagues on transcriptome data, that successfully separated GCB from ABC DLBCL[38]. Here, we applied LPS on methylome data, with a cohort composed of all 58 RS samples, 215 CLLs, and 68 DLBCLs. As we aimed to best discriminate between CLL and DLBCL profiles, only the highly differential CpGs between the two groups were considered in the analysis (261,085; FDR < 0.01; moderated *t*-statistics were retained for further use in the score computation). To lessen the impact of B cell IGHV maturity on the scoring model, we next subtracted CpGs that were also differential between U-CLL and M-CLL (128,408; FDR < 0.01). The 181,231 remaining CpGs were then filtered into 4863 CpGs with high methylation differential (beta-value differential or $\beta$-Fold-Change), that is, >30%. This amount was considered appropriate as: (i) statistical power to discriminate such a methylation differential was reached; (ii) probe composition was balanced between regulatory region/gene body/intergenic location as compared to the background; (iii) it provided sufficient number to expect a normal distribution of LPS within subgroups; and (iv) those CpGs demonstrated a strong correlation structure among the groups of samples (Fig. 2e).

Finally, from each of these 4863 CpGs and for each sample *S* of the cohort, the score

$$\text{LPS}(S) = \sum_{i=1}^{n} t_i . S_i \qquad (1)$$

was calculated, with $t_i$ representing the moderated *t*-statistic for CpG *i* and $S_i$ the corresponding methylation $\beta$-value. Known score distribution of CLL and DLBCL samples within their respective subgroup $G \in$ [CLL,DLBCL] allowed the Bayesian likelihood approximation for RS samples *S* to belong in each one of them, with probability

$$P(S \text{ in } G = \text{CLL}) = \frac{\Phi\left(\text{LPS}(S), \hat{\mu}_{\text{CLL}}, \hat{\sigma}^2{}_{\text{CLL}}\right)}{\Phi\left(\text{LPS}(S), \hat{\mu}_{\text{CLL}}, \hat{\sigma}^2{}_{\text{CLL}}\right) + \Phi\left(\text{LPS}(S), \hat{\mu}_{\text{DLBCL}}, \hat{\sigma}^2{}_{\text{DLBCL}}\right)} \qquad (2)$$

and $P(S \text{ in } G = \text{CLL}) \simeq 1 - P(S \text{ in } G = \text{DLBCL})$ where $\Phi$ computes the normal density function with the estimated means $\hat{\mu}$ and variances $\hat{\sigma}^2$ of LPS within either subgroup *G*. To finally obtain highly specific and homogeneous subgroups, thresholds were defined as follows: (i) $p(S \text{ in } G = \text{CLL}) \geq 0.98$ for CLL-derived (namely $p_{\text{CLL-derived}}$), and (ii) $p(S \text{ in } G = \text{DLBCL}) \geq 0.98$ for DLBCL-like ($p_{\text{DLBCL-like}}$) labeling, since the gray zone between the two main groups is centered on scores for which the probability density functions overlap at values >0.02 either way (Supplementary Fig. 10).

## Transcriptomics

All samples were processed within the same batch. Demultiplexed single-end sequencing data corresponding to 50-nucleotide-long reads were available in FASTQ files, one for each of the 87 samples, and used in the next processing steps. The cohort was composed of 47 RS samples, 2 paired-CLLs, 28 DLBCLs, and 10 controls from normal lymph nodes. The 10 normal controls were added for methodologic purposes: (i) normalization; (ii) checking benign profiles against B cell malignancies; (iii) checking the feeble amplitude of the transcriptomic component separating inflammatory (*n* = 3) from non-inflammatory lymph nodes (*n* = 7); and (iv) validation of the efficiency of the developed scoring methods.

**Transcriptome reconstruction pipeline.** First quality controls were conducted using FastQC v0.11.5 [http://www.bioinformatics.babraham.ac.uk/projects/fastqc/] results as a guideline. No adapter content or known overrepresented sequence needed to be removed at this step. Read mapping and the main filtering were performed using HISAT2 v2.0.4[62] against a reference index built to account for human population SNPs as well as known transcripts (this index can be obtained from ftp://ftp.ccb.jhu.edu/pub/infphilo/hisat2/data/grch38_snp_tran.tar.gz). The following scoring constraints were applied during alignment:–score-min L, 0,–0.2-sp 10.3–dta. Samtools v1.3.1 [http://github.com/samtools/samtools] was used for manipulating the alignment files throughout the downstream analysis. PCR duplicates were flagged with Picard v1.13 MarkDuplicates [https://broadinstitute.github.io/picard/]. Differentially spliced transcripts were assembled from the obtained alignments with StringTie v2.1.0[63]. The present protocol took advantage of the proposed workflow for identifying known as well as novel isoforms, using an annotation file for hg38 in gtf format as a guide [ftp://ftp.ensembl.org/pub/release88/gtf/homo_sapiens/Homo_sapiens.GRCh38.90.gtf.gz]. The following parameters were used: (i) first step is applied for each sample -f 0.2 -j 3 -c 10 -M 0.5, (ii) second step merges all transcripts of all samples–merge -m 200 and (iii) the last step estimates abundances and read coverage for all merged transcripts, for each sample -A -C -f 0.2 -j 3 -c 10 -M 0.5. Two tables were generated from these results, one compiling raw read count at the gene level, and another at the transcript level.

**Raw abundance filtering and normalization.** Raw counts were filtered by applying a minimum expression threshold for a gene or transcript. Those had to be expressed (non-zero value) in at least two samples and present an average expression value across all samples higher than 1/5,000,000 of the average library size ($64 \pm 3$ million reads per sample), that is, at least 20 reads per feature. Data was further adjusted with the TMM normalization method[64], and finally was log2 and cpm (count per million) transformed[65]. A total of 23,508 genes and 77,491 transcripts were identified and reported at the end of the process. Pearson's correlations for gene expression levels averaged at 0.92 for genes and 0.75 for transcripts and were very stable across samples (data not shown).

**Gene and transcript annotations.** All transcriptomic analyses were performed using the hg38 reference assembly of the human genome. Results were fully annotated with known symbols corresponding to gene and transcript genomic locations whenever possible. Upon completion of the transcript assembly, gene symbols were assigned Ensembl IDs based on overlapping positions with known transcripts (90% overlap minimum). In case of failed overlap, custom and unique IDs were used. Therefore, gene and transcript assignments were based on the Ensembl[66] GRCh38 annotations available in both *core* and *funcgene* databases, version 90. These were downloaded from [ftp://ftp.ensembl.org/pub/release-90/mysql/](ftp://ftp.ensembl.org/pub/release-90/mysql/) for local installation and query with in-house custom tools).

**Transcriptome explorations.** Unsupervised analyses were all carried out with hierarchical and *K*-means clustering techniques, as previously described[67]. Expression values were median-centered, and uncentered Pearson's correlation was used as distance metrics. Supervised analyses were performed through linear modeling (empirical Bayes), and differential expression *p* values were obtained using a two-way moderated *t* test then adjusted for FDR following the Benjamini−Hochberg procedure. An FDR < 0.01 indicated statistical significance. Cluster dissection was achieved with functional annotation tools for target gene associations, such as the Open Targets platform[68], and gene signature correlation with public datasets from multiple databases, such as GEO (Gene Expression Omnibus), with Enrichr [https://maayanlab.cloud/Enrichr].

**Methylome and transcriptome data integrations**
Here we focused on the RS cohort, for which 41 RS samples overlapped between methylome and transcriptome experiments. A subset of the methylome EPIC dataset (M-values, normalized and curated) and part of the transcriptome dataset (gene and transcript CPMs – also normalized and filtered) were integrated to eliminate unwanted signals and pinpoint the functional mechanisms linking DNA methylation of regulatory regions with gene expression in RS.

Both datasets were re-annotated with biomaRt[69] and linked using two methods: (i) with shared Ensembl identifiers; and (ii) by genomic coordinates for refined feature overlap when the first method failed. We used "TSS200," "TSS1500," and "first exon" CpG information to define associations with promoter regions in the next analysis steps, and overlap was considered successful within 2 kb between CpG and gene transcription start sites (TSS). The integromes generated at this step represented 475,148 and 674,567 associations at the gene and transcript levels, respectively. As described in a similar setup[70], Spearman's correlations were calculated for each association. Correlations at the gene level were used for generating density plots and presenting a general view, whereas transcripts were used for precise analyses and final results. These were filtered into candidate transcriptional effector locations, by selecting "promoter regions" containing at least three negatively-correlated CpGs (rho < −1/3; *p* value <0.001) or three positively correlated CpGs (rho > 1/3; *p* value <0.001) with features corresponding to "TSS200," "TSS1500," "first exon," or "TSSoverlap2kb" (each linked to the same transcript identifier).

Manhattan representations were plotted against the background with the R CMplot package. Gene set enrichment and pathway analyses of selected candidate lists were carried out as described in *Methylome data analyses*. Interaction networks of putative TFs encoded by candidate genes, protein domain enrichments, and effector functions were performed with STRING tools [https://string-db.org/][71]. A curated database of 1639 human TFs with DNA-binding domain information was obtained from http://humantfs.ccbr.utoronto.ca/. Regulatory networks were built with NetworkAnalyst [www.networkanalyst.ca][72].

**Methodology for building the gene expression-based scoring system**
**CLL-derived RS signature.** A 215-gene set was obtained by extracting two clusters of strongly correlated up- and down-regulated profiles from the transcriptome hierarchical clustering tree (Fig. 3a, Supplementary Fig. 36, and Supplementary Data 9). The two initial clusters displayed a very high enrichment in CLL genes and mainly drove the whole sample aggregation process. These were further reduced to protein-coding genes, to avoid biases when applying the signature to transcriptomes of different origins, which may not contain ncRNAs or genes of undefined biotype. The reduced set was then overlapped with genes integrating significantly between transcriptome and methylome. The resulting 215-gene signature contained 93 protein-coding genes underexpressed in CLL-derived RS and 122 protein-coding genes overexpressed in CLL-derived RS.

**Linear classifier score (LCS).** For each analyzed dataset, scores were obtained according to the following procedure, to render the process as reproducible as possible. (i) When applicable, raw expression data with relevant sample annotations were retrieved from the Gene Expression Omnibus curated database (https://www.ncbi.nlm.nih.gov/geo/) with GEOquery[73]. Expression matrices were then prepared, described statistically, and normalized according to a well-established protocol[74]. Otherwise, already normalized expression data were used "as is". (ii) Whole transcriptomes were reduced to their features (genes, transcripts, probes) corresponding to matches with the 215-gene signature. (iii) Expression values were summed up over genes to obtain an aggregated and unique expression for each gene. (iv) Data were scaled, i.e., mean-centered and standard-deviation-reduced. (v) Positive outliers were trimmed at the last permille (99.9%) to reduce the impact of extreme gene expression values on the score but preserve high enough values as essential markers. Trimmed values were replaced with the last permille value. After a distribution check, no negative outliers were found in any dataset. (vi) For each of the 215 genes, weights were assigned: those originating from the upregulated cluster were weighted +1 and those originating from the downregulated cluster were weighted −1. (vii) Finally, LCS scores were computed as the mean of weighted gene expressions for each sample *S* of the dataset:

$$LCS(S) = \frac{1}{n}\sum_{i=1}^{n} G_i \cdot W_i \qquad (3)$$

with *n* the number of genes in the signature, and $G_i$ representing the gene *i* weighted by $W_i$. LCS scores were then standardized (mean-centering to 0 and standard-deviation-reducing to obtain scores fully comparable between datasets). The obtained *Z*-scores were compared to a normal distribution in a one-way test to calculate a *p* value, used to define the initial LCS cutoff ($p < 0.05$) in each dataset.

**Statistics and reproducibility**
No statistical method was used to predetermine sample size. Data exclusion criteria according to quality controls are explained in the

"Methods" section. The experiments were not randomized. The investigators were not blinded to allocation during experiments and outcome assessment.

### Reporting summary

Further information on research design is available in the Nature Portfolio Reporting Summary linked to this article.

## Data availability

Raw DNA methylation, gene expression and targeted NGS data generated in this study from RS samples have been deposited in the European Genome-Phenome Archive (study EGAS00001005495) under accession number EGAD00010002194 for DNA methylation data; accession number EGAD00001007922 for transcriptomic data, and accession number EGAD00001009509 for targeted NGS data. The raw data are protected and available under restricted access. Clinical and genomic data can be obtained by contacting the data access committee, according to the European Genome-Phenome Archive's procedure. Data access will be granted if their use complies with the data use conditions, including a commitment to strictly use these data for a clearly identified academic research programs and according to good practice recommendations. The Data Access Committee will respond to requests within 2 weeks. Once access to the data is granted, these are available until the end of the research program they support. Previously published DNA methylation datasets from the ICGC MMML-seq consortium that were used in this study are available upon request from the data access committee at the ICGC consortium data portal [https://dcc.icgc.org/]. Published datasets can be found under the following accession codes: GSE103265; GSE66770; GSE10846; GSE98588; GSE87371. All other data supporting the findings of this study are available from the corresponding authors upon request. Source data are provided with this paper.

## Code availability

The source code developed for this study for designing the DNam and gene expression classifiers and the methylome–transcriptome integrative analyses is available on the GitHub platform, [https://github.com/zetcheuv/RichterOmicsCode]. All other source data supporting the findings of this study are available from the corresponding authors.

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

## Acknowledgements

The authors would like to thank the divisions of clinical hematology, hematology laboratory and pathology of Nancy (Dr Hélène Busby, Pr Hervé Sartelet, Dr Ludovic Dubouis), Poitiers, Angers, Reims (Dr Pascale Cornillet-Lefebvre), Clermont-Ferrand (Dr Lauren Veronèse and Dr Albane Ledoux-Pilon), Tours (Dr Flavie Arbion), Avicenne, Saint-Louis (Dr Véronique Meignin) and Pitié-Salpêtrière (Dr Frédéric Charlotte and Pr Isabelle Brocheriou). The authors would like to thank the tumor libraries biological resource centers of Nancy (BB-0033-00035), Poitiers (BB-0033-00068), Caen (Pr Xavier Troussard), Tours, Clermont-Ferrand, Angers (BB -0033-00038), Reims-Champagne-Ardenne, Besançon (Franck Monnien, Dr Etienne Daguindau) and Bordeaux (Marie-Pierre Fort, Dr Fontanet Bijou) who provided us with the biological material. The authors would like to thank Véronique Saunier (direction of research at University Hospital of Nancy) for supporting the project. RNA sequencing was performed by the GenomEast platform, a member of the "France Génomique" consortium (ANR–10-INBS-0009). The authors would like to thank Louis Staudt (Center for Cancer Genomics, National Cancer Institute, Bethesda, MD 20892, USA) for giving access to the data published in Schmitz and colleagues (2018) and Wright and colleagues (2020). The authors would like to thank the members of ICGC the MMML-seq consortium for contribution to the generation of the DLBCL omics datasets and the MMML-seq consortium for data access. The authors would like to thank Pr Catherine Wu and Dr Erin Parry for shared expertise and language editing, and Dr Cath Carsberg for English language editing. This work was supported in part by the Cancéropôle Est (J.B., S.H., P.F.), the Ligue contre le Cancer (J.B., P.F.), the University Hospital of Nancy (J.B., P.F.), the association of SILLC patients (J.B., P.F.), and the Association des Chefs de Services of the University Hospital of Nancy (J.B.). E.T., S.S., and R.S. were supported by the DFG (SFB1074 projects B1, B9, and B10). The ICGC MMML-Seq consortium has been supported by the German Ministry of Science and Education in the framework of the ICGC MMML-Seq consortium (01KU1002) and ICGC DE-Mining (01KU1505).

## Author contributions

Conception and design: J.B., S.H., P.F., R.S., S.S. Development and methodology: S.H., J.B., E.T., M.K., P.F., J.I.M.-S., R.S., S.S. Sample and clinical data providing, acquired and managed patients: C.D., D.R.W., A.Q., O.B., C. Tomowiak, G.L., G.G., S.L., E.C., F.N.K., F.D., A.R., M.-C.B., A.D., O.T., G.O., M.H., C. Thieblemont, R.G., J.I.M.-S. and F.C. Acquisition of data (data production and techniques, provided facilities): J.B., S.H., J.V., M.K., R.H., P.R., C.C., D.M., E.C., C.S., S.L., E.T., S.B., G.O., J.-L.G., ICGC MMML-seq consortium, MMML consortium, P.F., R.S., S.S. Analysis and interpretation of data (e.g., statistical analysis, biostatistics, computational analysis): S.H., J.B., C.M., C.S., A.M., M.K., R.S., S.S. Writing, review and/or revision of the manuscript: J.B., S.H., P.F., R.S. and S.S. wrote the first and the revised version of the paper. All authors critically reviewed and agreed on the final version of the manuscript. Administrative, technical and material support (i.e., reporting or organizing data, constructing databases): S.H., J.B., J.V., C.M., E.C., S.L., E.T., P.L., O.A., ICGC MMML-seq consortium, MMML consortium, P.F., R.S., S.S.

## Competing interests

The authors declare no competing interests.

## Additional information

[1]Division of CLL. Department of Internal Medicine III, Ulm University, Ulm, Germany. [2]Inserm UMRS1256 Nutrition-Génétique et Exposition aux Risques Environnementaux (N-GERE), Université de Lorraine, Nancy, France. [3]Université de Lorraine, CHRU-Nancy, Service d'Hématologie Biologique, Pôle Laboratoires, F54000 Nancy, France. [4]Institute of Human Genetics, Ulm University & Ulm University Medical Center, Ulm, Germany. [5]Fraunhofer Institute for Cell Therapy and Immunology IZI, Leipzig, Germany. [6]Department of Haematology, University Hospital of Tours, Tours, France. [7]Department of Hematology, Hôpital de la Pitié-Salpêtrière, AP-HP, Paris, France. [8]Université de Reims Champagne-Ardenne, IRMAIC, Centre Hospitalier Universitaire de Reims, Hématologie Clinique, Reims, France. [9]Department of Hematology, University Hospital of Nancy, Vandoeuvre-lès-Nancy, France. [10]Inserm, CHRU, University of Lorraine, CIC Clinical Epidemiology, Nancy, France. [11]Department of Clinical Pathology, Robert-Bosch-Krankenhaus, and Dr. Margarete Fischer-Bosch Institute for Clinical Pharmacology, Stuttgart, Germany. [12]CHU Angers, Biological Resource Center of Angers (CRB-CHU Angers), BB-0033-00038, Laboratoire d'Hématologie, Angers, France. [13]Department of Hematology, CHU Poitiers, Poitiers, France. [14]CIC1402 Inserm Poitiers, Poitiers, France. [15]Hematology Laboratory, Avicenne Hospital, Assistance Publique-Hôpitaux de Paris, Paris, France. [16]Bioinformatics Group, Department of Computer Science and Interdisciplinary Center for Bioinformatics, Leipzig University, Leipzig, Germany. [17]Hematology department, Clermont-Ferrand University Hospital, Clermont-Ferrand, France. [18]Department of Biopathology CHRU-ICL, BBB, CHRU Nancy, Vandoeuvre-lès-Nancy, France. [19]Biological Resource Center of Nancy, BB-0033-00035, CHRU de Nancy, Nancy, France. [20]Sorbonne Université, Cytogénétique Hématologique, Hôpital Pitié-Salpêtrière, AP-HP, Paris, France. [21]Centre de Recherche des Cordeliers, INSERM, Université Sorbonne Paris Cité, Université Paris Descartes, Université Paris Diderot, F-75006 Paris, France. [22]CHRU of Nancy, Service de Biochimie-Biologie Moléculaire-Nutrition, Pôle Laboratoires, F54000 Nancy, France. [23]Hematology Department, Hôpital Pitié-Salpêtrière,

AP-HP, Sorbonne University, Paris, France. [24]Department of Hematology, University Hospital of Angers, Angers, France. [25]Institute of Pathology, University Hospital of Würzburg, Bavaria, Germany. [26]Hematology Biology, University Hospital of Nantes, Hôtel-Dieu, France. [27]Inserm 1232 Centre de Recherche en Cancérologie et Immunologie Nantes Angers (CRCINA), Nantes, France. [28]Department of Hematology, Hôpital Saint-Louis, Paris, France. [29]Division of Molecular Genetics, German Cancer Consortium (DKTK) and National Center for Tumor Diseases (NCT) Heidelberg, German Cancer Research Center (DKFZ), Heidelberg, Germany. [30]Biomedical Epigenomics Group, Institut d'investigacions Biomèdiques August Pi I Sunyer (IDIBAPS), University of Barcelona, Barcelona, Spain. [31]Institució Catalana de Recerca i Estudis Avançats (ICREA), Barcelona, Spain. [32]These authors contributed equally: Julien Broséus, Sébastien Hergalant. [33]These authors jointly supervised this work: Pierre Feugier, Reiner Siebert, Stephan Stilgenbauer. ✉e-mail: julien.broseus@univ-lorraine.fr; Stephan.Stilgenbauer@uniklinik-ulm.de

## ICGC MMML-Seq Consortium

Ole Ammerpohl[4], Stephan Bernhart[16], Markus Kreuz[5], Peter Lichter[29], German Ott[11], Andreas Rosenwald[25], Reiner Siebert[4,33] & Stephan Stilgenbauer [1,33]✉

A full list of members and their affiliations appears in the Supplementary Information.

