## [Peer review file · Nature Communications]

REVIEWER COMMENTS

Reviewer #1 (Remarks to the Author): Expert in lymphoma genomics and epigenomics

Patients with chronic lymphatic leukemia (CLL), an indolent lymphoma, can transform to aggressive B-cell lymphoma, mainly diffuse large B-cell lymphoma (DLBCL) in a process called Richter transformation (RT). The manuscript by Broseus and Hergalant et al performs an integrative analysis of DNA methylation data with transcriptomics from 58 samples of patients with RT and perform a comparative analysis to other major B-cell-derived malignancies, including CLL and primary DLBCL. Roughly 78% of patients shared a clonal relationship between their two B-cell diseases suggesting a linear evolution, while 22% of patients were clonally unrelated, largely confirming prior studies. Their study also confirmed hypomethylation in RT samples and their integrative analyses highlighted deregulated pathways and signaling intermediates associated with RT. The authors also constructed a molecular classifier for CLL-RS to identify RT with a clonal relationship to the CLL, which is an open question in the field. Using multidimensionally reduction technologies the authors identify that a minor fraction of RT patients is close to primary DLBCL which prompted them in the last section of the manuscript to project the signature of these tumors to primary DLBCL. Using this approach, they identify a small subset of DLBCL patients that had an inferior outcome.

Overall this is an interesting paper that integrates methylation and transcriptomic data in RT, a disease with a dismal prognosis. There are, however, several aspects that should improve and clarify the paper:

- 1.) Several aspects of the study are known and to this end, some of the aspects are confirmative. Hypomethylation in RT-DLBCL, Roughly an 80/20 split of clonally related to clonally not-related are known aspects of RT biology. The authors should emphasize their new findings and state confirmative results citing all prior work. Overall one wonders, which new aspect of RT their study highlighted for the biology of RT. Is there any new biological insight that can be used to treat this disease?
- 2.) RT without clonal relationship to the preexisting CLL has a better response rate to R-CHOP-based therapies compared to RT with a clonal relationship. To this end, it is interesting to define a classifier that can distinguish these two states (branched vs. linear evolution). However, classifiers are prone to overtraining when applied only to one dataset. In particular, when dealing with classifiers that might guide treatment decisions, one wonders how good is the proposed classifier in an independent dataset? In addition, the authors need to provide the gene signature in the supplement.
- 3.) The most difficult part seems to be the selection of a gene signature and applying it to primary DLBCL and claiming this as a new DLBCL subset with an inferior prognosis. At this moment in time, there are a plethora of signatures (Imaging, DNA-, RNA-, and protein-based) that are all defining inferior prognosis

in DLBCL. For some aspects the authors perform a multi-variate analysis, but what about others. The most obvious aspect would be an association to TP53 and/or MYC/BCL2 double or single hit, which has been linked to inferior prognosis and is reported to be enriched in relapsed DLBCL (as their signature). There needs to be a very systematic multivariate analysis of all known risk factors before the authors can claim a new DLBCL subgroup. An enrichment to CD5 positive DLBCL, which also has been associated with inferior outcomes, could also likely suggest that this is just a re-discovery of something described before. Again, the reviewer could not find the 214 gene list that confers these inferior groups, can this be added to the supplement? What are the biological features of this “new” subset? Do the authors claim that these are misdiagnosed RT?

4.) The authors need to make their data publicly available via one of the standard databases and provide an access number.

5.) Minor:

- a. At some points, the paper becomes a bit technical and descriptive, for instance, listing results of pathway enrichment without putting these pathways into context for the audience.
- b. Some figure labels, such as your Manhattan plot have labels of genes below visibility.
- c. Methods section could provide more detail.

Reviewer #2 (Remarks to the Author): Expert in RS and CLL genetics and clinical research

The work by Broseus and colleagues deals with a multi-omics characterization of Richter Syndrome (RS) cases that integrates data from DNA methylation, gene expression and targeted NGS.

Results show that DNA methylation is an efficient tool in differentiating DLBCL arising after or simultaneously to CLL from transformation of CLL to DLBCL.

In addition, the authors identify a novel DLBCL subgroup termed RS-type DLBCL.

I found this paper interesting, novel and with the merit of analyzing a really large cohort of RS cases.

Major issues:

1) The first one concerns the figures and their organization. Figure 2 for example, is made of 7 (A-G) panels, each occupying at least half a page, many a full page. How can the authors expect these panels to make up a single figure? A figure is something the reader (whether in this working format or in a printed paper) perceives as one delivering information that has to be physically together. I don't think the 7 panels of Figure 2 can fit in a single printed page. The authors should organise their figures in the appropriate way, proposing their idea of how it should be printed together, as it is normally done.

2) RS patient selection. Little is told about the 58 RS patients enrolled in the study, aside from the fact that their LN biopsies were reviewed by two pathologists. Did they all undergo FDG-PET/CT? Where were the LN obtained from? Were DNA/RNA extracted from frozen biopsies? And if so, how was tumor purity >70 established?

In addition, more information on the patients and their diagnosis would be important: I would suggest adding a table detailing clinical features of the enrolled patients.

It would be interesting to know, for example, if RS samples clustering with DLBCL were clinically distinguishable from other RS.

Lastly, the cohort is stated to be composed of 58 RS patients, but then gene expression is performed on 41 of these 58 plus "additional 7". What are these additional 7? Why was DNAm not performed on them?

3) DLBCL patient selection. The same considerations made above apply for this patient category. How was diagnosis made? Were DLBCL patients sub-classified into ABC or GC categories? Or in the 5 DLBCL subsets more recently identified?

4) A little more effort in explaining the results should be made. For example, line 149 to 154 are dedicated to explaining Figure 2C, panel RS vs CLL. The phrasing of the Results is identical to the Figure legend and it is unclear the point the authors want to make in commentating the differences. In this context, I would consider giving Figure S9 the status of a principal figure, given the message it conveys.

5) The statement from lines 168 to 172 should be integrated with clinical data. Are these two subgroups clinically different?

6) In devising their LPS score, the authors use CpGs that are non-influential in discriminating on the IGHV differentials CpGs. What is the rationale behind this? From stylistic point of view and to facilitate understanding by the reader, Figure 2D and Figure 3A should be rotated 90° to the right.

7) Figure 5 is unclear: how can the authors imagine having this figure as a figure? There are too many panels and the meaning is lost.

Reviewer #3 (Remarks to the Author): Expert in leukaemia genomics and methylation signatures

The manuscript by Broséus et al, describes a multi-omic analysis of a rare RS patient cohort. Herein, the authors use computational methods to integrate epigenomic and transcriptomic data, where they describe two molecularly distinct groups of RS, and further identify a subgroup of RS-similar DLBCL with poor outcome. The paper is generally well written and convincing, however there are several areas for improvement.

Major comments:

1. An overarching suggestion is for English language review and harmonization of terminology throughout. This is a large and complex study; more precise and careful wording would help improve the readability.

2. Batch effects: The authors make strong statements throughout the manuscript about these effects, and the methods applied to successfully correct for them. Therefore, it would be expected that they would more clearly point to or show the data backing up their claims.

a. There is no reference in the main text to the plots or other values to illustrate how well the batch effects (and tumor content effects) were corrected for. For example, figures S26-29 are not referred to in the main text.

b. Figure S27 is not readable in its present form.

c. Lines 122-137, "Data quality controls" only DNAm data are shown here, it would be good to mention the RNA-seq data as well. It was unclear if all RNA-seq data were prepared and processed in the same batch or not. M&M it seems to indicate all RNA-seq data were processed together, however in several places it is stated samples are "added" but no information explains whether these were data from previous work or newly generated data.

3. In its current form, the manuscript lacks cohesiveness in the way samples are explained/presented:

- a. The slight differences in how sample types are presented (referred to) throughout each section makes it hard for the reader to orient to which samples the authors are referring to. A suggestion to make it easier to follow is to include a table with the patient cohort(s); including additional information, like age range and other relevant clinical data, including tumor content of the samples (mean, med, range, etc).
- b. Information on what type of biopsy the DNA/RNA was extracted from. Was the same sample/biopsy used for DNAm analysis and RNA-seq?
- c. Since this is a study with several different datasets, it would be useful to have the number of samples/patients per dataset (i.e. methylation data, transcriptomic data).
- d. Could figure 1 be replaced by such a table? Or Figure 1 could be improved to reflect this.

4. In relation to comment 2, the concept of “paired samples” in general, and paired-CLLs (CLL-RS follow-ups?) in particular is extremely unclear. Are there repeated measurements from the same patient at different time points, or are all samples from distinct patients, or are these referring to the technical replicates on the different array platforms (EPIC, 450k)? For example, in figure S5, “paired RS” and “paired CLLs” are mentioned, but this it is quite unclear what these samples are. Nowhere else in the manuscript describes “paired RS”.

5. Line 149-156, a huge number of CpG sites were found to be differentially methylated between RS and CLL (~102k) and between RS and DLBCL (82k), so more than >30% of CpGs analyzed by the arrays (if this was based on CpGs present on both 450k and EPIC). What test was used to determine differential methylation, what were the cut-offs, how many samples in each group?

6. Given that hypomethylation of RS has already been established by others (ie ref 21), could this part of the results be slimmed down to give more room for the latter more novel aspects?

7. Figure 2:

- Panel A: Have the authors considered apply non-linear dimensionality methods like UMAP or t-SNE instead of PCA?

- Panel D: The heatmap in cannot properly distinguish these two subgroups as their methylation levels look quite similar. In line 183 you referred to these samples as ‘unclassified’ and put them in a separate group, but is it really a separate group? The hierarchical clustering on the Y axis is so squished that it is impossible to see the cluster organization. It appears some paired samples (or are these technical replicates?) included in the plot. Can this be indicated somehow? Out of curiosity, what does the methylation pattern of these 4k CpG sites look like in normal control samples?

- Panel F: This circos plot does not do a great job of indicating what the authors are trying to present here, ie genomic distribution of methylation values. Consider removing or moving to the supplement.

- Panel G: it would be much easier to read this plot if the box plots were labeled on the x-axis. What does “general methylation levels” actually refer to? From the boxplots it seems that the low and high score CLL-derived groups do not differ significantly in terms of general methylation levels; is it right to tell them apart as two separate groups?

8. High CLL-derived RS and Low CLL-derived RS. I am a little skeptical to this partitioning of the dataset. Can the authors explain why they feel this is relevant and sound solution? To me, it feels like they are removing the samples that do not agree with their hypothesis by lumping them into a separate group (ie low CLL-derived RS) and then proceeding to only analyze the samples with high scores? Is there a risk that the low CLL-derived RS samples are simply of lower tumor purity? Or something else? The gene expression data seems to do a better job of separating these two groups, what could be the reason?

9. The discussion is less well written than the rest of the manuscript, it in particular would benefit from editing and English language review. There are many long rambling sentences that are difficult to digest, for example lines on 374-381.

10. Can the code be made available? This would significantly improve the usability and reproducibility of this study.

Minor:

1. Line 68, bad-prognosis -> poor-prognosis

2. Figure 1 legend, unclear what “25 CLLs paired with RS” means.

3. Line 842- consider changing the word “regroup”.

4. Line 481-484: Why was a higher threshold of tumor content applied for RNA-seq (>70%) samples than DNAm samples (>50%). It seems counter intuitive as RNA-seq has a higher dynamic range and therefore (presumptively) would be more robust in the case of lower tumor cell content?

5. Fig 2C is very difficult to read, can the results be depicted in a different way?

6. Fig 3B, change color scale from Red/green.

7. Figure S5iii, missing x axis labels.

8. All figures, inconsistent usage of “RS”, “Richter”

9. Details missing in lines 312-316: 1) “showed very high DNAm and expression scores” <- what was “very high”? and 2) “with a statistically significant score” <- what was the value/test?

10. Line 388 of supplemental methods, reference out of format (Kulis)

Point-by-point response

Reviewer #1

Overall this is an interesting paper that integrates methylation and transcriptomic data in RT, a disease with a dismal prognosis. There are, however, several aspects that should improve and clarify the paper:

Q1 Major comment	Several aspects of the study are known and to this end, some of the aspects are confirmative. Hypomethylation in RT-DLBCL, Roughly an 80/20 split of clonally related to clonally not-related are known aspects of RT biology. The authors should emphasize their new findings and state confirmative results citing all prior work. Overall one wonders, which new aspect of RT their study highlighted for the biology of RT. Is there any new biological insight that can be used to treat this disease?
Response	We appreciate the opportunity to articulate the innovative aspects of our study. Indeed, we have uncovered new aspects of Richter syndrome (RS) biology by combining DNA methylation and gene expression profiling. Our novel findings, which have both biological and therapeutic implications, include: (i) The comprehensive profiling of RS genome-wide hypomethylation (Figure 2 [shown here])  Figure 2. DNA methylation comparative analysis with CLLs and de novo DLBCLs shows that Richter syndrome is a heterogeneous and hypomethylated entity. a) Unsupervised principal component analysis (PCA) of 58 RS, 215 CLLs, and 68 DLBCLs using the adjusted DNAm values of 397,769 QC-selected CpGs shared by

both 450K and EPIC microarray platforms; **b)** Global DNAm levels show a global hypomethylation in RS as compared with CLLs (U-CLL and M-CLL), DLBCLs, and subgroups of normal B-cells; **c)** Distribution of differential CpGs (FDR < 0.01; methylation differential > 10%) according to the reported chromatin states in 7 CLL reference epigenomes ⁽²⁰⁾; **d)** RS vs CLL top over-represented annotations, as returned by ReactomePA from 238 differential DMRs computed with DMRcate; **e)** DNA methylation-based linear predictor score (LPS) CpG architecture; **f)** Density map of DNAm between *high*CLL-derived and DLBCL-like RS. **g)** Boxplots showing general methylation levels for *high*CLL-derived, *low*CLL-derived, and DLBCL-like RS groups, together with *de novo* DLBCLs and CLLs.

- (ii) Dissection of the clonal vs non-clonal profiles in DNA methylation and gene expression
- (iii) The discovery of a DNA methylation CLL imprint, allowing clonal relationship assessment without the need of tumor DNA at CLL stage, thus implementing the panel of diagnostic tools
- (iv) Epigenetic architecture remodeling and subsequent deregulation of the EZH2 pathway, MGMT and PI3kinase/AKT and IGFR1 signaling cascades, all potentially actionable targets.
- (v) The central role of TP53, FOXC1 and NF-KB (**Figure S21**, shown here).

Figure S21: Regulatory network inferred from the set of 156 transcription factors correlating between methylome and transcriptome. These TFs are used as seeds for reconstructing the network (red dots, 1st line). Upstream, they are controlled by three master regulators (TP53, NFKB1 and FOXC1, blue dots). In turn, they regulate a large set of downstream TF effectors (green dots; 2nd line). TF interactions were computed from ENCODE and JASPAR databases.

- (vi) The discovery of a particular DLBCL subgroup associated with a RS-like profile and a poor prognosis.

We also identified an enrichment in the N1 profile in DLBCL sharing the CLL-derived Richter transcriptomic signature (**Figure 6**), indicating that a significant proportion of RS patients may benefit from Ibrutinib + R-CHOP therapy (Ref #43: Wilson *et al. Cancer Cell* 2021).

Figure 6. The gene expression-based linear classifier score (LCS) linearly classifies de novo DLBCL samples, with high scores enriched in N1, unclassified genomic profiles, and shorter progression-free survival.

Q2 Major comment

RT without clonal relationship to the preexisting CLL has a better response rate to R-CHOP-based therapies compared to RT with a clonal relationship. To this end, it is interesting to define a classifier that can distinguish these two states (branched vs. linear evolution). However, classifiers are prone to overtraining when applied only to one dataset. In particular, when dealing with classifiers that might guide treatment decisions, one wonders how good is the proposed classifier in an independent dataset? In addition, the authors need to provide the gene signature in the supplement.

Response

We thank the Reviewer for this valuable comment, and for this revision we have incorporated new analyses to address this question:

- First, our DNA methylation-based predictor is already trained with CLL and DLBCL cohorts, so overtraining is highly unlikely when we feed it with RS samples to classify them. It can provide a probability for a sample to be clonally related or unrelated according to a threshold.
- Second, we already identified a publicly available dataset (GSE103265), which we processed with hierarchical clustering (**Figure S22**) conducted on the 215-gene signature (**Table S9**). This clearly discriminated EBV-positive DLBCLs (primary DLBCLs) from EBV-negative DLBCLs (RS samples, which cluster with CLL). This strategy demonstrates that the gene signature is efficient for separating RS and CLL from DLBCLs, without using the complete classifier, but with a similar approach.

Figure S22: Validation of the CLL-derived RS gene signature on a DLBCL cohort including RS samples. From the public dataset GSE103265 available from the Gene Expression Omnibus database.

- Aside from this, we did not find other available public RS datasets, leading us to gather as many new RS samples as possible from France and Germany to build up an independent validation cohort, and process them using an EPIC 850K DNA methylation array, then submit these to our DNA methylation-based classifier.

Eighteen new RS samples (fresh frozen biopsies) were used to perform DNA methylation explorations using the EPIC 850K array. The CLL component was available for 14/18 (77.7%), allowing for clonal relationship assessment using the *IGHV*-based reference method. In addition, 6 new DLBCL samples, 6 new CLL samples and 4 RS samples from the training series (3 clonally related and 1 clonally unrelated, with respective CLL components) were used as internal controls. This new experiment enabled us to expand our investigation to 22 RS, 6 new DLBCLs, and 24 CLLs, including 18 paired CLLs.

Five of twenty-two (22.7%) RS samples were classified as DLBCL-like RS according to the DNA methylation-based linear predictor score, 3 of which were clonally unrelated to the CLL component (the CLL component was unavailable for the two other cases). The other 17 RS samples were classified as CLL-derived RS, with *IGHV*-assessed clonal relationship for 15/15 samples with concomitant CLL. This experiment was confirmative for all 8 samples already explored in the first EPIC 850K experiment.

At the end of the process, we confirmed that our DNAm-based classifier is able to delineate RS cases related to the preceding CLL, with a 100% match with the *IGHV*-based reference method (**Figure S16**).

Figure S16. hierarchical clustering of DNAm data retrieved from the validation cohort. This 52-sample validation cohort included 44 new samples: 18 new RS samples, the CLL component of 14 of these, 6 new DLBCL samples, and 6 additional CLLs. In addition, 8 samples from the training series were used as controls: 4 RS samples (3 clonally related and 1 clonally unrelated), with the 4 respective CLL components. RS sample classification according to linear predictor score (LPS) is displayed in the “RS subgroup” annotations.

Q3 Major comment	The most difficult part seems to be the selection of a gene signature and applying it to primary DLBCL and claiming this as a new DLBCL subset with an inferior prognosis. At this moment in time, there are a plethora of signatures (Imaging, DNA-, RNA-, and protein-based) that are all defining inferior prognosis in DLBCL. For some aspects, the authors perform a multi-variate analysis, but what about others. The most obvious aspect would be an association to TP53 and/or MYC/BCL2 double or single hit, which has been linked to inferior prognosis and is reported to be enriched in relapsed DLBCL (as their signature). There needs to be a very systematic multivariate analysis of all known risk factors before the authors can claim a new DLBCL subgroup. An enrichment to CD5 positive DLBCL, which also has been associated with inferior outcomes, could also likely suggest that this is just a re-discovery of something described before. Again, the reviewer could not find the 214 gene list that confers these inferior groups, can this be added to the supplement? What are the biological features of this “new” subset? Do the authors claim that these are misdiagnosed RT?
Response	We thank the reviewer for this constructive comment. Up to now, we have shown that gene expression-based linear classifier score (LCS) is a prognostic factor, independently from: GCB/ABC subtype and IPI (4 risk groups).  As DLBCL cohorts are external datasets, we can only work with available data. We repeated the enrichment analysis that we already performed in the first version of the manuscript: for MYC/BCL2 double hits, IPI, TP53 and all available variables, in every dataset (data on MYC/BCL2 double hit are available for Schmitz/Wright dataset only). In doing so, we demonstrated the

absence of enrichment in *MYC/BCL2* double hit, adverse IPI or *TP53* abnormalities in high linear classifier scores.

- Secondly, as suggested, we have now conducted a systematic multivariate analysis with a Cox Proportional Hazards model, including all informative covariates at hand to evaluate the gene expression-based linear classifier score vs survival (OS and PFS). We calculated the association of linear classifier score with survival in binary (top 25% linear classifier scores versus the rest) as well as linear (linear classifier score as a continuous variable) modes, provided estimates, and effect size for each covariate, as reported in **Figure S29**. The complete Cox model includes corrections for all possible covariates (as listed before). This multivariate analysis confirmed strong associations between linear classifier score and survival, independently of the effect and association levels of other covariates.

Cox PH multivariate statistics for PFS, variable = LCS, covariates = IPI + TP53 + DoubleHit (LR, Wald & logrank tests all < 1e-7)

	Beta	HR	P-Value	95% CI [LL, UL]	
LCS	0,4421	1,556	8,16E-05	[1.2488,1.939]	***
IPIcategory	0,7628	2,1442	2,11E-06	[1.5645,2.939]	***
TP53	0,41	1,5068	0,275	[0.7218,3.146]	
DoubleHit	-0,19	0,8269	0,569	[0.4296,1.592]	

Cox PH multivariate statistics for OS, variable = LCS, covariates = IPI + TP53 + DoubleHit (LR, Wald & logrank tests all < 9e-7)

	Beta	HR	P-Value	95% CI [LL, UL]	
LCS	0,36538	1,44106	0,00338	[1.1287,1.84]	**
IPIcategory	0,8426	2,32241	9,26E-07	[1.6587,3.252]	***
TP53	0,01682	1,01696	0,96862	[0.44,2.351]	
DoubleHit	-0,23064	0,79402	0,52805	[0.3879,1.625]	

Cox PH multivariate statistics for PFS, variable = top25, covariates = IPI + TP53 + DoubleHit (LR, Wald & logrank tests all < 2e-6)

	Beta	HR	P-Value	95% CI [LL, UL]	
top25	0,7127	2,0395	0,00327	[1.2683,3.279]	**
IPIcategory	0,7805	2,1826	8,68E-07	[1.5993,2.979]	***
TP53	0,3648	1,4402	0,33169	[0.6895,3.008]	
DoubleHit	-0,1739	0,8404	0,59978	[0.4389,1.609]	

Cox PH multivariate statistics for OS, variable = top25, covariates = IPI + TP53 + DoubleHit (LR, Wald & logrank tests all < 5e-6)

	Beta	HR	P-Value	95% CI [LL, UL]	
top25	0,587363	1,799238	0,0265	[1.0708,3.023]	*
IPIcategory	0,861689	2,367155	4,48E-07	[1.694,3.308]	***
TP53	0,007843	1,007874	0,9854	[0.4353,2.334]	
DoubleHit	-0,212922	0,808219	0,5566	[0.3974,1.644]	

Figure S29. Multivariate analysis with a Cox proportional hazards model, including all available informative covariates to evaluate the association of linear classifier score with survival (OS and PFS). This association was calculated in binary (top 25% linear classifier score versus the rest, tagged linear classifier score in the tables) as well as linear (linear classifier score as a continuous variable, tagged Zscore in the table) modes. Cox proportional-hazards multivariate models: (i) PFS model includes linear classifier score, IPI, TP53 abnormalities, and MYC/BCL2 double hit (Wald = 1e-8); (ii) OS model includes linear classifier score, IPI, TP53 abnormalities and MYC/BCL2 double hit (Wald = 5e-7). CI: confidence interval, HR: hazard ratio;

	LCS: linear classifier score; LL: lower limit; LR: likelihood ratio; UL: upper limit. * :p-value < 0.05; ** : p-value < 0.01; ***: p-value < 0.001.  • CD5+ enrichment is not the only characteristic of these RS-like DLBCLs as they also feature: (i) strong enrichment in ABC-subtype; (ii) strong enrichment in N1 and genomically unclassified DLBCLs (Wright et al. 2020), (iii) a central role of TP53, FOXC1 and NF-KB, delimiting a very special subgroup that has not been described before. We do not claim that these extreme phenotypes are misdiagnosed RS, but rather a particular DLBCL subgroup, with poor prognosis. • The 215-gene list was previously provided in the originally submitted version of the manuscript in Table S10 (Table S9 in the revised version).
Q4 Major comment	The authors need to make their data publicly available via one of the standard databases and provide an access number.
Response	We completely agree that the data we have generated should be publicly available. At the time of the original submission, the RS DNA methylation and transcriptome datasets were already recorded in EGA (European Genome-phenome archive) academic database, with accession numbers. This was already stated, along with dataset IDs, in the Methods section. To improve clarity, we have now moved this information into a newly created “Data and code availability” paragraph in the Methods section, with accession numbers and accession hyperlinks: Data and code availability statement Genomic data of RS samples were all newly generated. DNA methylation and transcriptome datasets from RS samples are accessible on the European Genome-Phenome Archive: study EGAS00001005495; accession number EGAD00010002194 for DNA methylation data; accession number EGAD00001007922 for transcriptomic data. Code developed for this study is available on the GitHub platform, at the following link: https://github.com/zetcheuv/RichterOmicsCode. EGA is among the public repositories accepted by and listed as recommended by the Nature publishing group. As these data were generated from human samples, allowing potential identification of patients based on genomic data, we are responsible for the reasonable use of these data. According to European laws, these sensitive data are accessible under conditions (including commitment to use these data according to good practice recommendations), with Data Access Committee policies.
Q5 Minor comment	At some points, the paper becomes a bit technical and descriptive, for instance, listing results of pathway enrichment without putting these pathways into context for the audience.
Response	We thank the reviewer for this suggestion, and have revised the manuscript accordingly. We partially rewrote the results section to make it more accessible to the readership, particularly for pathway enrichments. We also reinstated some methods into main text to improve clarity.
Q6 Minor comment	Some figure labels, such as your Manhattan plot have labels of genes below visibility.
Response	We thank the reviewer for this suggestion. New versions of Manhattan plots and other figures where gene names and features appear (Fig 2C, Fig 4B, Fig 4C , and shown below) were produced to increase the size of gene labels and make reading easier.

2C

4B

4C

Q7 Minor comment	Methods section could provide more detail.
Response	We appreciate the Reviewer's constructive comment. We have thus now we moved some details regarding the methods from the supplementals to the main text to improve clarity.

Reviewer #2

The work by Broseus and colleagues deals with a multi-omics characterization of Richter Syndrome (RS) cases that integrates data from DNA methylation, gene expression and targeted NGS. Results show that DNA methylation is an efficient tool in differentiating DLBCL arising after or simultaneously to CLL from transformation of CLL to DLBCL. In addition, the authors identify a novel DLBCL subgroup termed RS-type DLBCL. I found this paper interesting, novel and with the merit of analyzing a really large cohort of RS cases.

Q1 Major comment	The first one concerns the figures and their organization. Figure 2 for example, is made of 7 (A-G) panels, each occupying at least half a page, many a full page. How can the authors expect these panels to make up a single figure? A figure is something the reader (whether in this working format or in a printed paper) perceives as one delivering information that has to be physically together. I don't think the 7 panels of Figure 2 can fit in a single printed page. The authors should organize their figures in the appropriate way, proposing their idea of how it should be printed together, as it is normally done.
Response	We thank the reviewer for this very relevant comment, as some of these panels are indeed too large. • Combining 6-8 panels into a single figure is frequent in reference papers in the field (Kulis et al. Nature Genetics 2015; Kretzmer et al. Nature Genetics 2015; Beekman et al. Nature Medicine 2018). While each figure is important and displays a part of the message, we aimed at reducing them to the more informative panels, improving figure readability while preserving the main messages. Figure 2F (in the first version) is the biggest panel and needs to be displayed at a large scale to be fully readable. We have thus moved this figure into the supplementals, where it will be displayed as a full figure (Figure S14).• We lightened Figure 2E (now 2F) by removing the right part (ie half the figure) as it only explains how to read the left part, which is in fact a classical, easily accessible way of presenting data. We also partially lightened and clarified Figure 2C, and rewrote its legend to make it more easily readable and more accessible. Doing so left us room to display Figures 2A, 2C and 2E in full format while Figures 2B, 2F and 2G are now displayed at a smaller size.• According to the reviewer's comment #4, we also added the previous Figure S9 (now Figure 2D), to the new version of the Figure 2, which now fits onto a single page, as the majority of panels may be displayed at a small size. The updated and revised Figure 2 is now displayed here, below:

Figure 2. DNA methylation comparative analysis with CLLs and de novo DLBCLs shows that Richter syndrome is a heterogeneous and hypomethylated entity. **a)** Unsupervised principal component analysis (PCA) of 58 RS, 215 CLLs, and 68 DLBCLs using the adjusted DNAm values of 397,769 QC-selected CpGs shared by both 450K and EPIC microarray platforms; **b)** Global DNAm levels show a global hypomethylation in RS as compared with CLLs (U-CLL and M-CLL), DLBCLs, and subgroups of normal B-cells; **c)** Distribution of differential CpGs (FDR < 0.01; methylation differential > 10%) according to the reported chromatin states in 7 CLL reference epigenomes ⁽²⁰⁾; **d)** RS vs CLL top over-represented annotations, as returned by ReactomePA from 238 differential DMRs computed with DMRcate; **e)** DNA methylation-based linear predictor score (LPS) CpG architecture; **f)** Density map of DNAm between *high*CLL-derived and DLBCL-like RS. **g)** Boxplots showing general methylation levels for *high*CLL-derived, *low*CLL-derived, and DLBCL-like RS groups, together with de novo DLBCLs and CLLs.

Q2 Major comment

RS patient selection. Little is told about the 58 RS patients enrolled in the study, aside from the fact that their LN biopsies were reviewed by two pathologists. Did they all undergo FDG-PET/CT? Where were the LN obtained from? Were DNA/RNA extracted from frozen biopsies? And if so, how was tumor purity >70 established? In addition, more information on the patients and their diagnosis would be important: I would suggest adding a table detailing clinical features of the enrolled patients. It would be interesting to know, for example, if RS samples clustering with DLBCL were clinically distinguishable from other RS. Lastly, the cohort is stated to be composed of 58 RS patients, but then gene expression is performed on 41 of these 58 plus "additional 7". What are these additional 7? Why was DNAm not performed on them?

Response

We thank the Reviewer for these absolutely relevant comments, which we have now fully addressed, as described below:

- Clinical and biological data of the Richter cohort were in fact described in the original version of the paper, although in the Supplemental data (**Table S5**). Given the importance of this data, we have now revised the organization of the data so that it is provided as a new **Table 1**. In addition, the cohort is split

into CLL-derived and DLBCL-like Richters to show their respective characteristics.

- Most of our RS patients benefited from FDG-PET/CT at diagnosis and at follow-up (55/58; among the 3 missing: 2 CLL-derived and 1 DLBCL-like). These 58 patients originate from a larger cohort, described in a previous work from our group (*Ref #45: Moulin et al. AM J Hematol 2021*). This information has been now added to the main text in the Methods section ('Patients and Materials').
- The LN cases originated from 9 FILO centers in the framework of a national multicenter trial (ClinicalTrials.gov Identifier: NCT03619512). DNA/RNA were exclusively extracted from fresh frozen biopsies, tumor purity assessment was centrally performed by two pathologists (this information is now described in the Methods section).
- Regarding genomic data that we generated for this RS cohort, as RS is a rare disease, we had to conduct a retrospective study to maximize the cohort size. To improve quantity and quality, DNA and RNA extractions were performed separately with specific kits, and not with a dual kit (see Methods section, 'DNA and RNA extraction and qualification').
- Since RNA-seq is quite sensitive to contamination by cells from other cell populations, we were quite stringent with sample selection for RNA-seq and chose to apply a threshold of a minimum of 70% large cells in the parent specimen. In contrast, we found a threshold of 50% large cells as sufficient for DNA methylation, since our downstream bioinformatic protocols were able to correct the measurements according to cellular content. As a consequence, we had less samples eligible for RNA-seq. For these reasons, we were able to obtain DNA methylation data for 58 samples but matched RNA-seq data for only 41 cases.
- When frozen biopsies were too small for extracting both DNA and RNA, we prioritized gene expression data (provided that the large cell component was $\geq 70\%$). For a small subset of samples ($n=6$), only RNA-seq data were generated (see Material and Methods section [patients and materials] and the modified version of the **Figure 1**, shown below here).

Figure 1. Project Study workflow. Genome-wide DNA methylation data were available for 58 RS, 25 CLLs paired with RS (tumor DNA samples were available at both CLL and RS stages), 190 other CLLs, 68 de novo DLBCLs, and 92 samples from normal B-cells spanning the entire B lineage.

Q3
Major
comment

DLBCL patient selection. The same considerations made above apply for this patient category. How was diagnosis made? Were DLBCL patients subclassified into ABC or GC categories? Or in the 5 DLBCL subsets more recently identified?

Response	The 68 DLBCL cases used as reference group in this study originate from a larger lymphoma cohort built by the MMML (Molecular Mechanisms in Malignant Lymphoma) in the context of the ICGC (International Cancer Genome Consortium) MMML-seq consortium and described in Ref #46 Hübschmann et al. Leukemia 2021. Briefly, diagnoses were made according to the WHO 2008 criteria, all tumor specimens were collected before treatment, in the framework of a multicenter study, DLBCLs were subclassified according to gene expression-based GCB/ABC profiles. The genomically determined DLBCL subtypes from Schmitz et al. 2018 and Wright et al. 2020 were not used here. This information has been now added to the updated version of the manuscript (please see Methods/patients material)																																																																																																																																																																																																											
Q4 Major comment	A little more effort in explaining the results should be made. For example, line 149 to 154 are dedicated to explaining Figure 2C, panel RS vs CLL. The phrasing of the Results is identical to the Figure legend and it is unclear the point the authors want to make in commenting the differences. In this context, I would consider giving Figure S9 the status of a principal figure, given the message it conveys. Reminder : line 149 to 154 “Next, we annotated differentially methylated CpGs between RS, CLL, and DLBCL according to 12 chromatin states reported in seven CLL reference epigenomes. The 102,614 CpGs differentially methylated between RS and CLL (90.8% hypomethylations in RS) were depleted (ratio < 0.75) in active promoters, poised promoters, promoter-associated strong enhancers and weak promoters and enriched (ratio > 1.5) in transcription transition regions, and heterochromatin (Figure 2C).”																																																																																																																																																																																																											
Response	We appreciate the opportunity to clarify our findings and their interpretation.  2C Enrichment score: 1/4 1 4 Relative methylation status in RS: unmethylated (light grey), intermediate (dark grey), methylated (black)    Chromatin state RS vs CLL RS vs DLBCL   U-CLL M-CLL U-CLL M-CLL U-CLL M-CLL U-CLL M-CLL U-CLL M-CLL     WkTxn 1.11.31.11.31.31.21.2 0.60.70.50.80.60.70.6   WkProm 0.40.60.20.50.50.20.2 1.11.30.81.31.11.61.2   WkEnh 1.41.41.21.41.40.91 1.61.51.61.71.321.5   TxTrans 1.61.61.51.71.51.61.6 21.61.71.91.31.91.6   TxnElong 0.910.91110.9 0.50.50.50.60.50.50.5   StrEnh2 1.41.61.41.61.61.41.2 1.91.81.92.11.81.91.5   StrEnh1 0.50.60.30.60.30.40.3 0.81.10.510.60.70.5   Poisprom 0.20.30.30.30.40.30.3 2.72.732.82.72.62.8   Het:LowSign 1.61.71.71.71.71.51.6 0.80.80.80.90.81.10.9   H3K9me3Repr 1.11.31.111.20.81 1.1111.30.91.40.8   H3K27me3Repr 11.21.11.21.40.70.8 2.11.821.81.42.52.5   ActProm 0.20.30.30.30.30.30.3 0.20.30.30.30.30.30.3     • As stated before (comment 1), we modified Figure 2C and improved corresponding legends and comments in the main text to make it more easily understandable. c) Distribution of differential CpGs (FDR < 0.01; methylation differential > 10%) according to the reported chromatin states in 7 CLL reference epigenomes.⁽²⁰⁾ Enrichments are shown as a heatmap (blue: below expectation; red: higher than expected) and were calculated from the position of the selected CpGs in 7 CLL reference epigenomes (5 M-CLLs and 2 U-CLLs). Their distribution was reported among 12 different chromatin state categories in the CLL epigenomes. Barplots in the right part of each panel show the methylation status difference (meth status diff) in RS versus CLL or DLBCL: differentially methylated CpGs are distributed among 3 methylation level categories, namely unmethylated (beta-value < 0.3; light grey), intermediate (beta-value in 0.3-0.7; dark grey) and methylated (beta-value > 0.7; black). Upward bars indicate a comparative gain of CpGs in RS for the corresponding category, while downward bars indicate a comparative loss in RS. For example, differential CpGs between RS and DLBCL show a strong and constant enrichment in “poised promoter” locations among all reference epigenomes. These CpGs are also hypomethylated in	Chromatin state	RS vs CLL						RS vs DLBCL						U-CLL	M-CLL	U-CLL	M-CLL	U-CLL	M-CLL	U-CLL	M-CLL	U-CLL	M-CLL	WkTxn	1.1	1.3	1.1	1.3	1.3	1.2	1.2	0.6	0.7	0.5	0.8	0.6	0.7	0.6	WkProm	0.4	0.6	0.2	0.5	0.5	0.2	0.2	1.1	1.3	0.8	1.3	1.1	1.6	1.2	WkEnh	1.4	1.4	1.2	1.4	1.4	0.9	1	1.6	1.5	1.6	1.7	1.3	2	1.5	TxTrans	1.6	1.6	1.5	1.7	1.5	1.6	1.6	2	1.6	1.7	1.9	1.3	1.9	1.6	TxnElong	0.9	1	0.9	1	1	1	0.9	0.5	0.5	0.5	0.6	0.5	0.5	0.5	StrEnh2	1.4	1.6	1.4	1.6	1.6	1.4	1.2	1.9	1.8	1.9	2.1	1.8	1.9	1.5	StrEnh1	0.5	0.6	0.3	0.6	0.3	0.4	0.3	0.8	1.1	0.5	1	0.6	0.7	0.5	Poisprom	0.2	0.3	0.3	0.3	0.4	0.3	0.3	2.7	2.7	3	2.8	2.7	2.6	2.8	Het:LowSign	1.6	1.7	1.7	1.7	1.7	1.5	1.6	0.8	0.8	0.8	0.9	0.8	1.1	0.9	H3K9me3Repr	1.1	1.3	1.1	1	1.2	0.8	1	1.1	1	1	1.3	0.9	1.4	0.8	H3K27me3Repr	1	1.2	1.1	1.2	1.4	0.7	0.8	2.1	1.8	2	1.8	1.4	2.5	2.5	ActProm	0.2	0.3	0.3	0.3	0.3	0.3	0.3	0.2	0.3	0.3	0.3	0.3	0.3	0.3
Chromatin state	RS vs CLL						RS vs DLBCL																																																																																																																																																																																																					
	U-CLL	M-CLL	U-CLL	M-CLL	U-CLL	M-CLL	U-CLL	M-CLL	U-CLL	M-CLL																																																																																																																																																																																																		
WkTxn	1.1	1.3	1.1	1.3	1.3	1.2	1.2	0.6	0.7	0.5	0.8	0.6	0.7	0.6																																																																																																																																																																																														
WkProm	0.4	0.6	0.2	0.5	0.5	0.2	0.2	1.1	1.3	0.8	1.3	1.1	1.6	1.2																																																																																																																																																																																														
WkEnh	1.4	1.4	1.2	1.4	1.4	0.9	1	1.6	1.5	1.6	1.7	1.3	2	1.5																																																																																																																																																																																														
TxTrans	1.6	1.6	1.5	1.7	1.5	1.6	1.6	2	1.6	1.7	1.9	1.3	1.9	1.6																																																																																																																																																																																														
TxnElong	0.9	1	0.9	1	1	1	0.9	0.5	0.5	0.5	0.6	0.5	0.5	0.5																																																																																																																																																																																														
StrEnh2	1.4	1.6	1.4	1.6	1.6	1.4	1.2	1.9	1.8	1.9	2.1	1.8	1.9	1.5																																																																																																																																																																																														
StrEnh1	0.5	0.6	0.3	0.6	0.3	0.4	0.3	0.8	1.1	0.5	1	0.6	0.7	0.5																																																																																																																																																																																														
Poisprom	0.2	0.3	0.3	0.3	0.4	0.3	0.3	2.7	2.7	3	2.8	2.7	2.6	2.8																																																																																																																																																																																														
Het:LowSign	1.6	1.7	1.7	1.7	1.7	1.5	1.6	0.8	0.8	0.8	0.9	0.8	1.1	0.9																																																																																																																																																																																														
H3K9me3Repr	1.1	1.3	1.1	1	1.2	0.8	1	1.1	1	1	1.3	0.9	1.4	0.8																																																																																																																																																																																														
H3K27me3Repr	1	1.2	1.1	1.2	1.4	0.7	0.8	2.1	1.8	2	1.8	1.4	2.5	2.5																																																																																																																																																																																														
ActProm	0.2	0.3	0.3	0.3	0.3	0.3	0.3	0.2	0.3	0.3	0.3	0.3	0.3	0.3																																																																																																																																																																																														

	RS, with a shift from intermediate methylation (loss) towards the unmethylated (gain) category, with no methylated CpG involved.  • We also think Figure S9 is important - thank you. Accordingly, we have now added it in the new version of the Figure 2 (panel 2D), which now fits in a single page.
Q5 Major comment	The statement from lines 168 to 172 should be integrated with clinical data. Are these two subgroups clinically different? Reminder: line 168 to 172: the second component in the PCA split RS samples into two subgroups, one with a profile similar to DLBCL, the other closer to CLL (Figure 2A). Therefore, we postulated that we could distinguish “CLL-derived RS” samples, maintaining a CLL imprint, from “DLBCL-like RS”, distinct from the preceding CLL and close to de novo DLBCLs.
Response	We certainly agree that this is absolutely relevant.  • Clinical and biological data of the Richter cohort were already described in the originally submitted version of the manuscript, although this information was placed in the supplementals (Table S5). In this table, the cohort is divided into (i) CLL-derived; and (ii) DLBCL-like Richter cases, and their respective characteristics are provided. This table has been now moved to the main text as Table 1. • The two groups are similar for the majority of these parameters, except for the number of treatment lines at CLL stage ($p=0.02$), overall survival ($p=1.7 \times 10^{-3}$), IGHV mutated cases proportion ($p=6.3 \times 10^{-9}$), proportion of clonally related cases ($p=5.8 \times 10^{-6}$), and MYD88 mutations ($p=7 \times 10^{-3}$).
Q6 Major comment	In devising their LPS score, the authors use CpGs that are non-influentials in discriminating on the IGHV differentials CpGs. What is the rationale behind this? From stylistic point of view and to facilitate understanding by the reader, Figure 2D and Figure 3A should be rotated 90° to the right.
Response	We thank the reviewer for giving us the opportunity to clarify this point.  • CpGs with methylation level highly dependent on IGHV mutational status were excluded from DNA methylation-based linear predictor score (LPS) to prevent the classifier from simply classifying CLL, de novo DLBCLs and RS into two groups, according to IGHV mutational status. Based on published analyses, this would likely have been the case if we had included corresponding CpGs (Kulis et al. Nat Genet 2012; Queiros et al. Leukemia 2015). • The statistical model used to pinpoint very differential CpGs could correct for IGHV status, but may still be indirectly strongly influenced by it because it is constructed by comparing CLL and de novo DLBCL groups. • Therefore we processed the data differently. De novo DLBCLs are all IGHV-mutated, but CLL is either IGHV mutated or unmutated. Accordingly, we completely removed CpGs highly differential according to IGHV status to focus on other differences between CLLs and de novo DLBCLs and built a robust scoring method relying on two different reference groups (CLLs and de novo DLBCLs) from a set of non-influenced CpGs in CLL (ie excluding those identified in Kulis et al. Nat Genet 2012; Queiros et al. Leukemia 2015). We modified the text (results section, paragraph “DNA methylation separates CLL-derived and DLBCL-like RS subgroups”). In this way, the clonal or non-clonal classification of RS samples would not be influenced by this important feature.

- To definitely validate this approach, we came back to the initial 397,769 QC-selected CpGs and applied UMAP (Uniform Manifold Approximation and Projection) method (**Figure S3**).

Figure S3. UMAP of all CpGs from FULL dataset. Left part: UMAP applied to the 58 RS, 215 CLL and 68 DLBCL cases, on 397,769 QC-selected CpGs shared by both 450K and EPIC microarray platforms. Right part: UMAP of CLL and *de novo* DLBCLs (top), with predicted scattering of RS cases (bottom), demonstrating that sample distribution is not driven by *IGHV* mutational status. CLL: chronic lymphocytic leukemia; DLBCL: *de novo* diffuse large B-cell lymphoma; M-RS: *IGHV*-mutated Richter syndrome; RS: Richter syndrome; U-RS: *IGHV*-unmutated Richter syndrome.

- This showed that the two reference groups (CLL and DLBCLs) were homogeneous and clearly separated one from another (meaning that M-CLLs did not cluster with *de novo* DLBCLs). Then, we entered the RS samples into the model and showed that their position was independent from *IGHV* mutational status, as some *IGHV* unmutated RS cluster within *de novo* DLBCLs.

In conclusion, removing CpGs highly influenced by *IGHV* mutational status from our DNAm-based scoring system and focusing on other distinctive biological features between CLL and *de novo* DLBCLs is methodologically relevant, particularly in the perspective of detecting a CLL DNA methylation imprint.

- Anyway, we did not make this CpG selection for the next PCA, with the 10,000 most variable CpGs (**Figure 3C**). This figure shows that disease partitioning prevails over *IGHV* mutational status, as *IGHV* mutated CLLs do not cluster within *de novo* DLBCLs (while also *IGHV* mutated)

Figure 3: RS gene expression profiles corroborate DNA methylation subgrouping. c) Sample partitioning according to *IGHV* mutational status. Unsupervised PCA

clustering of U-RS, M-RS, U-CLL, M-CLL, and DLBCL according to the 10,000 most variable CpGs in the dataset.

- In accordance with the Reviewer's suggestion, Figures 2D and 3A have been rotated 90° to the right, which we find improves their presentation.

Q7
Major comment
Figure 5 is unclear: how can the authors imagine having this figure as a figure? There are too many panels and the meaning is lost.

Response
We thank the Reviewer for this constructive critique. After consideration, we have moved the majority of the panels of this figure to the supplementals (**now Figures S26-28**), and now display only original figures 5D and 5E (**now Figures 5A and 5B**) as they convey the main message from the data.

Figure 5. DLBCLs harboring the CLL-derived RS epigenetic signature are associated with ABC phenotype and worse outcome.

a) Kaplan-Meier estimates of progression-free survival for 429 patients from three combined and clinically annotated public DLBCL datasets.⁽⁸⁻¹⁰⁾ Comparative PFS between patients with top linear classifier score (LCS) and the rest of the cohorts, according to COO. Statistical comparisons were performed with the log-rank test. Bonferroni method was used for multiteesting adjustments.

b) Kaplan-Meier estimates of overall survival for 780 patients from four combined and clinically annotated DLBCL public datasets.^(8-10, 36) Comparative OS between patients with top linear classifier score (LCS) and the rest of the cohorts, according to COO. Statistical comparisons were performed with the log-rank test. Bonferroni method was used for multiteesting adjustments.

Reviewer #3

The manuscript by Broséus et al, describes a multi-omic analysis of a rare RS patient cohort. Herein, the authors use computational methods to integrate epigenomic and transcriptomic data, where they describe two molecularly distinct groups of RS, and further identify a subgroup of RS-similar DLBCL with poor outcome. The paper is generally well written and convincing, however there are several areas for improvement.

Q1 Major comment	An overarching suggestion is for English language review and harmonization of terminology throughout. This is a large and complex study; more precise and careful wording would help improve the readability.
Response	We thank you for this constructive suggestion. Accordingly, we have reviewed the entire work with a professional English medical writer. In particular, terminology was harmonized throughout the text.
Q2 Major comment	Batch effects. The authors make strong statements throughout the manuscript about these effects, and the methods applied to successfully correct for them. Therefore, it would be expected that they would more clearly point to or show the data backing up their claims. a. There is no reference in the main text to the plots or other values to illustrate how well the batch effects (and tumor content effects) were corrected for. For example, figures S26-29 are not referred to in the main text. b. Figure S27 is not readable in its present form. c. Lines 122-137, "Data quality controls" only DNAm data are shown here, it would be good to mention the RNA-seq data as well. It was unclear if all RNA-seq data were prepared and processed in the same batch or not. M&M it seems to indicate all RNA-seq data were processed together, however in several places it is stated samples are "added" but no information explains whether these were data from previous work or newly generated data.
Response	As the Reviewer pertinently suggested: a. We referred to former Figures S26-29, now Figures S32-35 in the main text (methods section). b. Figure S27 (now Figure S32) has been modified, focusing on the replicate samples in every batch, with specific coloured dots. Sample names were removed and replaced by filled dots. Dots representing un-replicated samples were kept in gray in the background.

Figure S32. Multi-dimensional scaling before and after normalization of the FULL dataset. Color emphasis on the 7 *de novo* DLBCL replicates between French and German facilities, and 2 RS and 2 CLL replicates between French batches. Upper panel: before normalization, showing a small shift between many replicates, always in the same direction. Lower panel: after normalization, showing the correction of the shift, as seen here between replicate samples.

c. Detailed transcriptome pipeline is now better explained in the supplemental methods. Indeed, all RNA-seq data were processed in the same batch (material and methods section, paragraph “Library preparation and RNA-sequencing”). We also explicitly stated that these are newly generated data (methods section, “data availability statement”). Indeed, « added » is not the appropriate term here, and we apologize for confusion on this point. In fact, these are additional samples (ie without associated DNA methylation data) processed in the same RNA-seq batch.

Q3
Major
comment

In its current form, the manuscript lacks cohesiveness in the way samples are explained/presented:

- The slight differences in how sample types are presented (referred to) throughout each section makes it hard for the reader to orient to which samples the authors are referring to. A suggestion to make it easier to follow is to include a table with the patient cohort(s); including additional information, like age range and other relevant clinical data, including tumor content of the samples (mean, med, range, etc).
- Information on what type of biopsy the DNA/RNA was extracted from. Was the same sample/biopsy used for DNAm analysis and RNA-seq?
- Since this is a study with several different datasets, it would be useful to have the number of samples/patients per dataset (i.e. methylation data, transcriptomic data).
- Could figure 1 be replaced by such a table? Or Figure 1 could be improved to reflect this.

Response

We thank the reviewer for giving us the opportunity to clarify these points.

- Corresponding relevant clinical data were initially displayed in **Table S5**. As suggested, we moved this table to the main text (**now Table 1**). Figure 1 partially displays these information and we modified it and its comments to improve clarity. Tumor cell contents are displayed in **Table 1** and in the relevant figures (**2E, 3A** for example), showing that this is not a confounding variable and that sample distribution is not driven by this variable, which is

homogeneously distributed among the different subgroups. These patients originate from a larger cohort which is described in a previous work from our group (Ref#45, Moulin et al. *AM J Hematol* 2021). This has been added to the methods section (paragraph “patients and materials”).

b. DNA and RNA were extracted from diagnosis biopsy. Downstream DNA methylation and RNA-seq analysis were indeed conducted from the same sample/biopsy.

c. We added an additional part to **Figure 1** to make this point clearer for the readership.

d. We slightly modified **Figure 1** accordingly. Combined with the figure update mentioned above in c., it will improve clarity.

Figure 1. Project Study workflow. Genome-wide DNA methylation data were available for 58 RS, 25 CLLs paired with RS (tumor DNA samples were available at both CLL and RS stages), 190 other CLLs, 68 de novo DLBCLs, and 92 samples from normal B-cells spanning the entire B lineage.

Q4 Major comment	In relation to comment 2, the concept of “paired samples” in general, and paired-CLLs (CLL-RS follow-ups?) in particular is extremely unclear. Are there repeated measurements from the same patient at different time points, or are all samples from distinct patients, or are these referring to the technical replicates on the different array platforms (EPIC, 450k)? For example, in figure S5, “paired RS” and “paired CLLs” are mentioned, but this it is quite unclear what these samples are. Nowhere else in the manuscript describes “paired RS”.
Response	To clarify, these paired samples are repeated measurements from the same patient at different timepoints (CLL stage and RS stage). We stated this more clearly in the revised version of the manuscript (results section). We now use the classical term « paired-CLL » for CLL tumor samples coupled with the corresponding tumor sample at RS stage. The term « paired-RS » is no longer used.
Q5 Major comment	Line 149-156, a huge number of CpG sites were found to be differentially methylated between RS and CLL (~102k) and between RS and DLBCL (82k), so more than >30% of CpGs analyzed by the arrays (if this was based on CpGs present on both 450k and EPIC). What test was used to determine differential methylation, what were the cut-offs, how many samples in each group? Reminder: line 149 to 156 “Next, we annotated differentially methylated CpGs between RS, CLL, and DLBCL according to 12 chromatin states reported in seven CLL reference epigenomes. The 102,614 CpGs differentially methylated between RS and CLL (90.8% hypomethylations in RS) were depleted (ratio < 0.75) in active promoters, poised promoters, promoter-associated strong enhancers and weak promoters and enriched (ratio > 1.5) in transcription transition regions, and heterochromatin (Figure 2C). The 82,940 CpGs

	differentially methylated between RS and DLBCL (96.4% hypomethylations in RS) were depleted (ratio < 0.75) in active promoters and enriched (ratio > 1.5) in poised promoters and regions repressed by H3K27me3.”
Response	We acknowledge that this aspect should be better emphasized. It was explained in supplementary materials (paragraph downstream bioinformatics/supervised analyses). We have also added these informations in the main text: “two-way moderated t-test adjusted for a FDR < 0.01” Test: two-way moderated t-test, linear modelling with empirical Bayes. This is case/control or binary EWAS. Cut-off: p-values adjusted for a FDR < 0.01, following the Benjamini-Hochberg procedure. No cut-off of methylation differential is applied here, as this is an introducing general description. Further cut-off (>30% methylation change for example) are applied in the next steps. Sample size : entire dataset, ie RS (n=58), CLL (n=215), DLBCL (n=68)
Q6 Major comment	Given that hypomethylation of RS has already been established by others (ie ref 21), could this part of the results be slimmed down to give more room for the latter more novel aspects?
Response	This is an important point to discuss.  This aspect of RS has been described, however with a 27K array that covers only a very limited part of the genome, mainly proximal promoters. Here we used a genome-wide microarray and we think it is important to describe the hypomethylation state of RS beyond the proximal promoters, as it unleashes expression of a large number of genes. Then we confirmed the interesting results from Rinaldi et al. and also uncovered the genome-wide hypomethylation in RS, together with the specific mechanisms it unleashes (with CLL epigenome annotations, Figure 2C). The gain/losses in methylation status for each epigenetic state is also depicted and reported in Figure 2C. To avoid insisting too much on this aspect, we chose to move Figure 2F to the supplements (now Figure S14) and state clearly and quickly in the main text the message it conveys.

Figure S14. Genomic distribution of DNA methylation in *high*CLL-derived RS (brown), DLBCL-like RS (orange) and DLBCLs (grey). Circular plot of median methylation levels and differences over sliding windows of 500 kb. Highlighted tracks 3 and 5 show the DNAm changes between *high*CLL-derived, DLBCL-like RS and DLBCL, which are different in both extent and locations. From outer to inner track: 1) Chromosome number and ideograms with cytobands (X and Y chromosomes were removed from the analysis); 2) *high*CLL-derived RS methylation levels; 3) *high*CLL-derived minus DLBCL-like RS methylation differences (range -10 to +30%). Areas hypermethylated in DLBCL-like RS are represented with orange peaks pointing towards the DLBCL-like RS track; 4) DLBCL-like RS methylation levels; 5) DLBCL-like RS minus DLBCL methylation differences (range -19 to +7%). Areas hypermethylated in DLBCL-like RS are represented with orange peaks pointing toward the DLBCL-like RS track while areas hypermethylated in DLBCLs are represented with grey peaks pointing towards the DLBCL track; 6) DLBCL methylation levels. Dashed lines represent 0%, 25%, 50%, and 75% average methylation.

Q7	Figure 2
Major comment	Panel A: Have the authors considered apply non-linear dimensionality methods like UMAP or t-SNE instead of PCA? Panel D: The heatmap in cannot properly distinguish these two subgroups as their methylation levels look quite similar. In line 183 you referred to these samples as ‘unclassified’ and put them in a separate group, but is it really a separate group? The hierarchical clustering on the Y axis is so squished that it is impossible to see the cluster organization. It appears some paired samples (or are these technical replicates?) included in the plot. Can this be indicated somehow? Out of curiosity, what does the methylation pattern of these 4k CpG sites look like in normal control samples? Panel F: This circo plot does not do a great job of indicating what the authors are trying to present here, ie genomic distribution of methylation values. Consider removing or moving to the supplement. Panel G: it would be much easier to read this plot if the box plots were labeled on the x-axis. What does “general methylation levels” actually refer to? From the boxplots it seems that the low and high score CLL-derived groups do not differ significantly in terms of general methylation levels; is it right to tell them apart as two separate groups?
Response	As pertinently suggested:  • Panel A: we added an UMAP of all CpGs from FULL dataset as Figure S3. This method confirms the results from the PCA. We initially favored PCA because of robust results (UMAP is non-linear and will give different layout

each time). We thank the reviewer for this excellent remark about UMAP, as it allowed us to compute layout for CLLs and DLBCLs only, then inject RS cases into it and predict its spread. It improves our approach and help explain why we chose to remove highly influential *IGHV* CpGs from the scoring set. t-SNE is less appropriate here because resource-consuming and unusable on very high dimensional datasets (400k CpGs), it would require to focus on a small CpG set.

Figure S3. UMAP of all CpGs from FULL dataset. Left part: UMAP applied to the 58 RS, 215 CLL and 68 DLBCL cases, on 397,769 QC-selected CpGs shared by both 450K and EPIC microarray platforms. Right part: UMAP of CLL and de novo DLBCLs (top), with predicted scattering of RS cases (bottom), demonstrating that sample distribution is not driven by *IGHV* mutational status. CLL: chronic lymphocytic leukemia; DLBCL: de novo diffuse large B-cell lymphoma; M-RS: *IGHV*-mutated Richter syndrome; RS: Richter syndrome; U-RS: *IGHV*-unmutated Richter syndrome.

- Panel D: the term “unclassified” comes from the DNA methylation-based linear predictor score (LPS), which applies a very stringent cut-off. Hierarchical clustering shows that $_{high}$ CLL derived RS and primarily unclassified RS cluster together and share a CLL DNA methylation imprint. Then we termed them « $_{low}$ CLL-derived-RS » to reflect this. Importantly, tumor cell contents were similar between $_{high}$ CLL-derived RS and $_{low}$ CLL-derived RS. We chose to display this on heatmap **Figures 2E and 3A** (below) to be very clear that tumor content cannot explain in itself the $_{high}/_{low}$ CLL-derived dichotomy.

We also improved the display of the hierarchical clustering. The Y axis was enhanced in height, as suggested. Paired samples were gathered at different timepoints (ie CLL phase and RS phase) in the same patients (as described above, we explained it better in the revised version), and were depicted in an annotation track by a colour matching code. Indeed, this is interesting to display these 4,863 CpGs in the normal B-cells throughout B-cell development, which we did and provided as a supplemental figure (**Figure S13**).

Figure S13. Methylation status of the 4,863 CpGs used by the linear classifier score (LCS) in normal B-cells. BCP: B-Cell precursors; gcBC: germinal center B-cells; MBC: memory B-cells; MGZ: marginal zone; naiBC: naive B-cells; PC: plasma cells.

- Panel F: according to the reviewer’s comment, **figure 2F** was moved to the supplemental material.

	 Panel G: we improved the figure annotations, according to the Reviewer's comment. While they scored differently in the DNA methylation-based linear predictor score, we thereafter demonstrated that $high_{CLL}$- and low_{CLL}-derived may in fact be considered in the same CLL-imprinted RS group. We added a $p=ns$ over these two boxplots to reflect this. Q8 Major comment	$High_{CLL}$-derived RS and low_{CLL}-derived RS. I am a little skeptical to this partitioning of the dataset. Can the authors explain why they feel this is relevant and sound solution? To me, it feels like they are removing the samples that do not agree with their hypothesis by lumping them into a separate group (ie low_{CLL}-derived RS) and then proceeding to only analyze the samples with high scores? Is there a risk that the low_{CLL}-derived RS samples are simply of lower tumor purity? Or something else? The gene expression data seems to do a better job of separating these two groups, what could be the reason?
Response	We appreciate the opportunity to clarify our findings and their interpretation.  $High_{CLL}$ derived RS are samples classified as CLL-derived in both linear predictor score and hierarchical clustering (Figure 2E). Low_{CLL}-derived RS = samples showing a CLL imprint in hierarchical clustering, but with a $p(CLl)$ below stringent cutoff (98%). So the terms « high » and « low » only reflect DNA methylation-based linear predictor scoring, with a stringent cut-off. It does not indicate different tumor cell contents.   To be perfectly balanced in our manner of discriminating samples with our model, we followed the same rules on both [for $p(CLl)$ and $p(DLBCL)$] and attributed these labels accordingly. We then realized that this unclassified subgroup is in fact clinically and biologically similar to $high_{CLL}$-derived RS subgroup, as both groups display a CLL imprint. We finally concluded that both $high_{CLL}$-derived and low_{CLL}-derived subgroups should certainly be included into the larger CLL-imprint group. Along the way, we always used all the samples and never discarded any in any kind of experiment. On that note, we would rather discuss the threshold for the linear predictor score:  -If $p(CLl) < 0.02$, we have no CLL imprint, this is probably a clonally unrelated RS and we can classify it to the « DLBCL-like » group. -If $p(CLl) \geq 0.98$, we have a strong CLL imprint, these samples can be unequivocally classified within the CLL-derived RS.

	-If p(CLL) is between 0.02 and 0.98, we may consider that the most important is the CLL imprint, even minor, and classify them along with the ^{high}CLL-derived RS into a larger « CLL-derived RS ».  • As mentioned above, the ^{high}- and ^{low}CLL-derived profiles do not reflect tumor purity, as shown in Figures 2E and 3A. So the difference between ^{high}- and ^{low} CLL-derived RS may reflect different evolution patterns from the CLL clone, ^{high}CLL-derived RS keeping close to CLL, and ^{low}CLL-derived RS rather evolving toward a primary DLBCL profile (molecular-wise). • ^{High}CLL-derived RS were used as the most typical/representative group for CLL-imprinted RS samples in further combined methylome-transcriptome analyses for which we needed opposite groups to unravel typical features of each of them (CLL-derived versus DLBCL-like). The only time we did not include them was to perform over-representation and pathway analyses, because these are very sensitive and we needed to focus on the specific subgroups we were comparing.
Q9 Major comment	The discussion is less well written than the rest of the manuscript, it in particular would benefit from editing and English language review. There are many long rambling sentences that are difficult to digest, for example lines on 374-381. Reminder: line 374 to 381 Adjusting for the lack of appropriate human or animal models to study RS and functionally validate our findings, our research set on with primary human RS samples, which also holds promise for discoveries because based on the primary diseases under study. Results significance was supported by cross-validation of stringent orthogonal methods, as DNA methylation patterns is largely corroborated by transcriptome-sequencing data, in a completely independent manner. We carefully evaluated the influence of tumor cell content of our samples and lowered it by deconvoluting the methylation data according to B-cell contents.
Response	Language has been improved after review by a professional English medical writer. The discussion section has been extensively rebuilt.
Q10 Major comment	Can the code be made available? This would significantly improve the usability and reproducibility of this study.
Response	The code developed for this study is available on the GitHub platform, at the following link: https://github.com/zetcheuv/RichterOmicsCode. It includes: 1) methylome and transcriptome complete workflows, guidelines and pipelines to generate the datasets; 2) code for methylome/transcriptome data integrations; 3) code for statistical analyses and data exploration; and 4) code for calculating methylome and transcriptome scores.
Q11 Minor comment	Line 68, bad-prognosis -> poor-prognosis
Response	The change has been made
Q12 Minor comment	Figure 1 legend, unclear what “25 CLLs paired with RS” means.
Response	“CLLs paired with RS” should say that samples were available from the same patient at both the CLL and RS disease stages; hereafter “paired-CLLs”. This was added to the comments of the Figure 1 and in the main text in the results section.

Q13 Minor comment	Line 842- consider changing the word “regroup”.																																																																																																																																																																																																											
Response	Corresponding sentence was removed from the comments of the figure 2, as it is explained in the main text. We used the term “cluster”.																																																																																																																																																																																																											
Q14 Minor comment	Line 481-484: Why was a higher threshold of tumor content applied for RNA-seq (>70%) samples than DNAm samples (>50%). It seems counter intuitive as RNA-seq has a higher dynamic range and therefore (presumptively) would be more robust in the case of lower tumor cell content?																																																																																																																																																																																																											
Response	We appreciate this comment (please refer also to Q2 of Reviewer #2). The minimal tumor purity requirement was raised to 70% for RNA analysis, as RNA-seq represents gene expression in a cumulative way. Signal from residual normal cells strongly influences global gene expression, especially for transcripts with very low expression in tumor cells, but high expression in contaminating normal cells. This is difficult to separate and assign with bioinformatics protocols. In contrast, DNA methylation signal from residual normal cells has a lower influence on global measurement and is more accessible to correction by cellular deconvolution methods. This has been added to the revised version of the manuscript (methods/patients and materials).																																																																																																																																																																																																											
Q15 Minor comment	Fig 2C is very difficult to read, can the results be depicted in a different way?																																																																																																																																																																																																											
Response	We thank the reviewer for this comment. We modified Figure 2C and improved corresponding legends and comments to make it more easily understandable.  2C Enrichment score: 1/4 1 4 Relative methylation status in RS: unmethylated (light grey), intermediate (dark grey), methylated (black) Chromatin state    Chromatin state RS vs CLL RS vs DLBCL   U-CLL M-CLL M-CLL U-CLL M-CLL     WkTxn 1.11.31.11.31.31.21.2 0.60.70.50.80.60.70.6   WkProm 0.40.60.20.50.50.20.3 1.11.30.81.31.11.61.2   WkEnh 1.41.41.21.41.40.91 1.61.51.61.71.321.5   TxTrans 1.61.61.51.71.51.61.6 21.61.71.91.31.91.6   TxnElong 0.910.91110.9 0.50.50.50.60.50.50.5   StrEnh2 1.41.61.41.61.61.41.2 1.91.81.92.11.81.91.5   StrEnh1 0.50.60.30.60.30.40.3 0.81.10.510.60.70.5   Polsprom 0.20.30.30.30.40.30.3 2.72.732.82.72.62.8   Het;LowSign 1.61.71.71.71.71.51.6 0.80.80.80.90.81.10.9   H3K9me3Repr 1.11.31.111.20.81 1.1111.30.91.40.8   H3K27me3Repr 11.21.11.21.40.70.8 2.11.821.81.42.52.5   ActProm 0.10.10.10.10.10.10.1 0.20.30.10.30.10.20.3    Meth status diff  Figure 2. c) Distribution of differential CpGs (FDR < 0.01; methylation differential > 10%) according to the reported chromatin states in 7 CLL reference epigenomes.⁽²⁰⁾ Enrichments are shown as a heatmap (blue: below expectation; red: higher than expected) and were calculated from the position of the selected CpGs in 7 CLL reference epigenomes (5 M-CLLs and 2 U-CLLs). Their distribution was reported among 12 different chromatin state categories in the CLL epigenomes. Barplots in the right part of each panel show the methylation status difference (meth status diff) in RS versus CLL or DLBCL: differentially methylated CpGs are distributed among 3 methylation level categories, namely unmethylated (beta-value < 0.3; light grey), intermediate (beta-value in 0.3-0.7; dark grey) and methylated (beta-value > 0.7; black). Upward bars indicate a comparative gain of CpGs in RS for the corresponding category, while downward bars indicate a comparative loss in RS. For example, differential CpGs between RS and DLBCL show a strong and constant enrichment in	Chromatin state	RS vs CLL						RS vs DLBCL						U-CLL		M-CLL		M-CLL		U-CLL		M-CLL		WkTxn	1.1	1.3	1.1	1.3	1.3	1.2	1.2	0.6	0.7	0.5	0.8	0.6	0.7	0.6	WkProm	0.4	0.6	0.2	0.5	0.5	0.2	0.3	1.1	1.3	0.8	1.3	1.1	1.6	1.2	WkEnh	1.4	1.4	1.2	1.4	1.4	0.9	1	1.6	1.5	1.6	1.7	1.3	2	1.5	TxTrans	1.6	1.6	1.5	1.7	1.5	1.6	1.6	2	1.6	1.7	1.9	1.3	1.9	1.6	TxnElong	0.9	1	0.9	1	1	1	0.9	0.5	0.5	0.5	0.6	0.5	0.5	0.5	StrEnh2	1.4	1.6	1.4	1.6	1.6	1.4	1.2	1.9	1.8	1.9	2.1	1.8	1.9	1.5	StrEnh1	0.5	0.6	0.3	0.6	0.3	0.4	0.3	0.8	1.1	0.5	1	0.6	0.7	0.5	Polsprom	0.2	0.3	0.3	0.3	0.4	0.3	0.3	2.7	2.7	3	2.8	2.7	2.6	2.8	Het;LowSign	1.6	1.7	1.7	1.7	1.7	1.5	1.6	0.8	0.8	0.8	0.9	0.8	1.1	0.9	H3K9me3Repr	1.1	1.3	1.1	1	1.2	0.8	1	1.1	1	1	1.3	0.9	1.4	0.8	H3K27me3Repr	1	1.2	1.1	1.2	1.4	0.7	0.8	2.1	1.8	2	1.8	1.4	2.5	2.5	ActProm	0.1	0.1	0.1	0.1	0.1	0.1	0.1	0.2	0.3	0.1	0.3	0.1	0.2	0.3
Chromatin state	RS vs CLL						RS vs DLBCL																																																																																																																																																																																																					
	U-CLL		M-CLL		M-CLL		U-CLL		M-CLL																																																																																																																																																																																																			
WkTxn	1.1	1.3	1.1	1.3	1.3	1.2	1.2	0.6	0.7	0.5	0.8	0.6	0.7	0.6																																																																																																																																																																																														
WkProm	0.4	0.6	0.2	0.5	0.5	0.2	0.3	1.1	1.3	0.8	1.3	1.1	1.6	1.2																																																																																																																																																																																														
WkEnh	1.4	1.4	1.2	1.4	1.4	0.9	1	1.6	1.5	1.6	1.7	1.3	2	1.5																																																																																																																																																																																														
TxTrans	1.6	1.6	1.5	1.7	1.5	1.6	1.6	2	1.6	1.7	1.9	1.3	1.9	1.6																																																																																																																																																																																														
TxnElong	0.9	1	0.9	1	1	1	0.9	0.5	0.5	0.5	0.6	0.5	0.5	0.5																																																																																																																																																																																														
StrEnh2	1.4	1.6	1.4	1.6	1.6	1.4	1.2	1.9	1.8	1.9	2.1	1.8	1.9	1.5																																																																																																																																																																																														
StrEnh1	0.5	0.6	0.3	0.6	0.3	0.4	0.3	0.8	1.1	0.5	1	0.6	0.7	0.5																																																																																																																																																																																														
Polsprom	0.2	0.3	0.3	0.3	0.4	0.3	0.3	2.7	2.7	3	2.8	2.7	2.6	2.8																																																																																																																																																																																														
Het;LowSign	1.6	1.7	1.7	1.7	1.7	1.5	1.6	0.8	0.8	0.8	0.9	0.8	1.1	0.9																																																																																																																																																																																														
H3K9me3Repr	1.1	1.3	1.1	1	1.2	0.8	1	1.1	1	1	1.3	0.9	1.4	0.8																																																																																																																																																																																														
H3K27me3Repr	1	1.2	1.1	1.2	1.4	0.7	0.8	2.1	1.8	2	1.8	1.4	2.5	2.5																																																																																																																																																																																														
ActProm	0.1	0.1	0.1	0.1	0.1	0.1	0.1	0.2	0.3	0.1	0.3	0.1	0.2	0.3																																																																																																																																																																																														

	“poised promoter” locations among all reference epigenomes. These CpGs are also hypomethylated in RS, with a shift from intermediate methylation (loss) towards the unmethylated (gain) category, with no methylated CpG involved.
Q16 Minor comment	Fig 3B, change color scale from Red/green.
Response	We did not fully understand this comment, as Figure 3B is already in red/green, and the current color scale is at the optimal setup. Non-significant clusters are covered with a grey filter to better emphasize the two significant clusters (C1 and C6). If the comment was dedicated to figure 3A, we changed the color scale from blue/yellow to red/green, which better emphasizes the most important clusters. We chose to keep this version and apply the same color code to all transcriptomic heatmaps.
Q17 Minor comment	Figure S5iii, missing x axis labels.
Response	The x-axis labels are now visible in the revised version (now Figure S6).
Q18 Minor comment	All figures, inconsistent usage of “RS”, “Richter”
Response	We thank the reviewer for this relevant comment. We harmonized the figures with the use of “RS”.
Q19 Minor comment	Details missing in lines 312-316: 1) “showed very high DNAm and expression scores” <- what was “very high”? and 2) “with a statistically significant score” <- what was the value/test?
Response	1) These samples show extreme scores in both DNAm- and gene expression-based classifiers and were separated from the bulk of samples. We modified the text accordingly. These samples are labelled in red in the updated version of the figure (Figure S23, panel A).   Figure S23. a) Correlation between methylome and transcriptome scores (linear predictor score and linear classifier score, respectively) in the ICGC MMML-seq dataset (scatterplot, linear modelling fit with 5% confidence interval). Samples with extreme scores in both DNAm and gene expression are labeled in red. b) Linear classifier score densities for the ICGC MMML-seq dataset (n=94); c) Linear classifier score densities for MMML dataset (n=430). b) and c): Red curve: observed distribution

	(lowess-smoothed); Green curve: simulated/expected distribution from the real observations (lowess-smoothed). 2) The statistics and full construction methods supporting the linear predictor score (DNA methylation-based) and linear classifier score (gene expression-based) are explained in the Methods section, paragraphs “Downstream bioinformatics / Linear predictor score (LPS)” and “Methodology for building the gene expression-based scoring system / Linear classifier score (LCS)”.
Q20 Minor comment	Line 388 of supplemental methods, reference out of format (Kulis)
Response	The correction has been made.

REVIEWERS' COMMENTS

Reviewer #1 (Remarks to the Author):

The authors have addressed my concerns.

Reviewer #2 (Remarks to the Author):

The paper was thoroughly and carefully revised and I found it much improved in readability and much easier to follow.

I just have few minor issues.

1) Line 72: shouldn't it be "determines" instead of "determined".

2) Line 95: "no fully faithful human or animal experimental models of RS exist and there is limited knowledge about RS biologic features". This is not entirely true, as there are patient-derived xenograft models of RS (doi: 10.1158/0008-5472.CAN-17-4004), which remain fairly similar to the primary samples that have been used to study signaling pathways (as an example doi: 10.1182/blood.202008276) and drug responses (doi: 10.1182/blood.2020010187 and doi: 10.1182/blood.2020008404, among others) that should be quoted. In addition, a cell line derived from a RS patient was recently characterized (10.1016/j.neo.2020.11.010) by some of the same authors of this paper.

3) In the Discussion section, lines 390-396. More than detailing why their research is "expanding knowledge in several ways", the authors list why their research is technically superior. "Firstly...robust and proven methods...", "Secondly,...comprehensive analysis...rather than pinpointing a limited number of specific targets". "Thirdly,...large cohorts...which contrasts with previous work mostly focusing on the RS transformation process".

All these points are methodological and technical, but per se they do not expand knowledge. The authors rephrase this paragraph.

4) Again, in line 398 the authors mention the lack of "appropriate...models". Cell lines and patient-derived xenografts should be mentioned and why there are deemed inappropriate to functionally confirm their results.

Reviewer #3 (Remarks to the Author):

Thank you for addressing all my remarks. I am satisfied with your responses.

During the review process a similar paper was published:

Nadeu, F., Royo, R., Massoni-Badosa, R. et al. Detection of early seeding of Richter transformation in chronic lymphocytic leukemia. *Nat Med* 28, 1662–1671 (2022). <https://doi.org/10.1038/s41591-022-01927-8>

The paper by Nadeu et al should be referred to in the present manuscript. Additionally, adjustment of some aspects in the introduction and discussion of the current manuscript will be warranted, as Nadeu addresses a similar question as well as point out some of the unique/novel findings presented herein. This may give the authors an opportunity to put their results into a wider context.

Reviewer #1 (Remarks to the Author):

The authors have addressed my concerns.

Reviewer #2 (Remarks to the Author):

The paper was thoroughly and carefully revised and I found it much improved in readability and much easier to follow.

I just have few minor issues.

Minor comment 1	Line 72: shouldn't it be "determines" instead of "determined".
Response to minor comment 1	We appreciate this correction. The change has been made.
Minor comment 2	Line 95: "no fully faithful human or animal experimental models of RS exist and there is limited knowledge about RS biologic features".  - This is not entirely true, as there are patient-derived xenograft models of RS (doi: 10.1158/0008-5472.CAN-17-4004), which remain fairly similar to the primary samples that have been used to study signaling pathways (as an example doi: 10.1182/blood.2020008276) and drug responses (doi: 10.1182/blood.2020010187 and doi: 10.1182/blood.2020008404, among others) that should be quoted. - In addition, a cell line derived from a RS patient was recently characterized (10.1016/j.neo.2020.11.010) by some of the same authors of this paper.
Response to minor comment 2	We thank the Reviewer for this valuable comment. We have incorporated a sentence about these models, citing the 5 suggested references: "As compared to other lymphoid malignancies, the availability of in vitro or in vivo models to study RS is limited,⁽¹⁰⁻¹⁵⁾ and therefore our current knowledge on RS biology remains incomplete."
Minor comment 3	In the Discussion section, lines 390-396. More than detailing why their research is "expanding knowledge in several ways", the authors list why their research is technically superior. "Firstly...robust and proven methods...", "Secondly,...comprehensive analysis...rather than pinpointing a limited number of specific targets". "Thirdly,...large cohorts...which contrasts with previous work mostly focusing on the RS transformation process". All these points are methodological and technical, but per se they do not expand knowledge. The authors rephrase this paragraph.
Response to minor comment 3	We are grateful for this comment. We have modified our text, by replacing "Our research is expanding this knowledge in several ways" by "Here we used a different study design and methodological approach to expand this knowledge.", which better introduces the point on our methodological and technical processes.
Minor comment 4	Again, in line 398 the authors mention the lack of "appropriate...models". Cell lines and patient-derived xenografts should be mentioned and why there are deemed inappropriate to functionally confirm their results.

Response to minor comment 4	We thank the reviewer for this relevant comment. In vitro and in vivo xenograft models and Richter-derived cell lines are now cited in the discussion section. We also explain why we consider that studying large cohorts of primary human RS samples better catch the wide complexity of this disease: “Human-derived xenograft mouse models and cell lines were recently reported to study RS biology and test drug response.⁽¹¹⁻¹⁵⁾ However, the availability of these models is limited and they cannot recapitulate the full heterogeneity of RS, as they were generated from a limited number of tumor samples. Our approach using large cohorts of primary human RS samples and comparative tumor material also holds promise for discoveries and better characterize the wide RS epigenetic complexity.”
--

Reviewer #3 (Remarks to the Author): Thank you for addressing all my remarks. I am satisfied with your responses.

Minor comment 1	During the review process a similar paper was published: Nadeu, F., Royo, R., Massoni-Badosa, R. et al. Detection of early seeding of Richter transformation in chronic lymphocytic leukemia. Nat Med 28, 1662–1671 (2022). https://doi.org/10.1038/s41591-022-01927-8 The paper by Nadeu et al should be referred to in the present manuscript. Additionally, adjustment of some aspects in the introduction and discussion of the current manuscript will be warranted, as Nadeu addresses a similar question as well as point out some of the unique/novel findings presented herein. This may give the authors an opportunity to put their results into a wider context.
Response to minor comment 1	We thank the Reviewer for this valuable comment. Our study and the study by Nadeu and colleagues are from our perspective adding to each other, as they address different aspects. Nadeu and colleagues explored the CLL to RS transformation process over time, with whole genome, gene-expression, and chromatin explorations. In our study, we mainly characterized the RS epigenetic profile and its heterogeneity within the DLBCL landscape, and in the light of DNA methylation and gene-expression. As the work from Nadeu et al. is highly relevant to RS in general, we have incorporated several aspects of this work in the introduction and the discussion section of our manuscript:  - In the introduction section: “Remarkably, a recent report using multiome and single cell approaches in sequential CLL-RS samples describe that the increased molecular complexity of RS does not seem to be the consequence of clonal evolution over time but rather the selection of minute subclones present at diagnosis years before overt transformation (Nadeu et al. Nat Med 2022).” - In the discussion section: “Such global hypomethylation may in part reflect a more extensive proliferative history of the RS subclone, (Nadeu et al. Nature

	Medicine 2022) as measured by the epiCMT mitotic clock (Duran-Ferrer et al. Nature Cancer 2022), and later in the discussion section: “However, a recent single cell transcriptome analysis of sequential CLL-RS samples revealed that, as compared to the CLL cells, RS cells downregulate genes related to BCR signaling and upregulate those involved in oxidative phosphorylation,(Nadeu et al. Nature Medicine 2022) and therefore RS may be less sensitive to Ibrutinib.”
--	--